# Near-Optimal Edge Evaluation in Explicit Generalized Binomial Graphs

**Sanjiban Choudhury**
The Robotics Institute
Carnegie Mellon University
sanjiban@cmu.edu

**Shervin Javdani**
The Robotics Institute
Carnegie Mellon University
sjavdani@cmu.edu

**Siddhartha Srinivasa**
The Robotics Institute
Carnegie Mellon University
siddh@cs.cmu.edu

**Sebastian Scherer**
The Robotics Institute
Carnegie Mellon University
basti@cs.cmu.edu

## Abstract

Robotic motion-planning problems, such as a UAV flying fast in a partially-known environment or a robot arm moving around cluttered objects, require finding collision-free paths quickly. Typically, this is solved by constructing a graph, where vertices represent robot configurations and edges represent potentially valid movements of the robot between these configurations. The main computational bottlenecks are expensive edge evaluations to check for collisions. State of the art planning methods do not reason about the *optimal sequence of edges* to evaluate in order to find a collision free path quickly. In this paper, we do so by drawing a novel equivalence between motion planning and the Bayesian active learning paradigm of *decision region determination (DRD)*. Unfortunately, a straight application of existing methods requires computation exponential in the number of edges in a graph. We present BISECT, an efficient and near-optimal algorithm to solve the DRD problem when edges are independent Bernoulli random variables. By leveraging this property, we are able to significantly reduce computational complexity from exponential to linear in the number of edges. We show that BISECT outperforms several state of the art algorithms on a spectrum of planning problems for mobile robots, manipulators, and real flight data collected from a full scale helicopter. Open-source code and details can be found here: https://github.com/sanjibac/matlab_learning_collision_checking

## 1   Introduction

Motion planning, the task of computing collision-free motions for a robotic system from a start to a goal configuration, has a rich and varied history [23]. Up until now, the bulk of the prominent research has focused on the development of tractable planning algorithms with provable *worst-case performance guarantees* such as computational complexity [3], probabilistic completeness [24] or asymptotic optimality [20]. In contrast, analysis of the *expected performance* of these algorithms on the real world planning problems a robot encounters has received considerably less attention, primarily due to the lack of standardized datasets or robotic platforms. However, recent advances in affordable sensors and actuators have enabled mass deployment of robots that navigate, interact and collect real data. This motivates us to examine the following question: "How can we design planning algorithms that, subject to on-board computation constraints, maximize their expected performance on the actual distribution of problems that a robot encounters?"

This paper addresses a class of robotic motion planning problems where *path evaluation is expensive*. For example, in robot arm planning [12], evaluation requires expensive geometric intersection

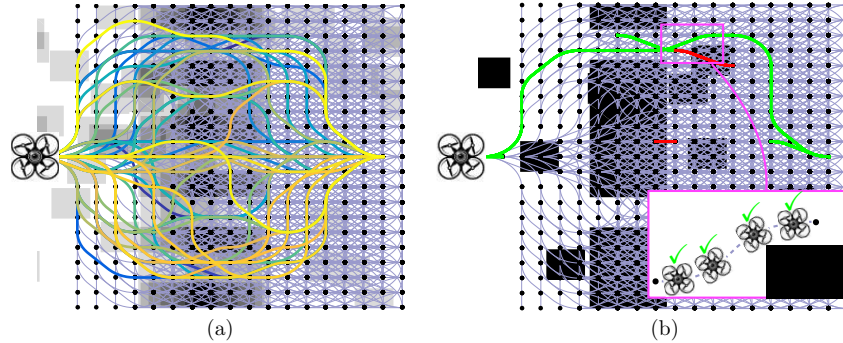

<div align="center">(a)                       (b)</div>

Figure 1: The feasible path identification problem (a) The explicit graph contains dynamically feasible maneuvers [27] for a UAV flying fast, with a set candidate paths. The map shows the distribution of edge validity for the graph. (b) Given a distribution over edges, our algorithm checks an edge, marks it as invalid (red) or valid (green), and updates its belief. We continue until a feasible path is identified as free. We aim to minimize the number of expensive edge evaluations.

computations. In UAV path planning [9], evaluation must be done online with limited computational resources (Fig. 1).

State of the art planning algorithms [11] first compute a set of unevaluated paths quickly, and then evaluate them sequentially to find a valid path. Oftentimes, candidate paths share common edges. Hence, evaluation of a small number of edges can provide information about the validity of many candidate paths simultaneously. Methods that check paths sequentially, however, do not reason about these common edges.

This leads us naturally to the *feasible path identification* problem - given a library of candidate paths, identify a valid path while minimizing the cost of edge evaluations. We assume access to a prior distribution over edge validity, which encodes how obstacles are distributed in the environment (Fig. 1(a)). As we evaluate edges and observe outcomes, the uncertainty of a candidate path collapses.

Our first key insight is that this problem is equivalent to *decision region determination (DRD)* [19, 5]) - given a set of tests (edges), hypotheses (validity of edges), and regions (paths), the objective is to drive uncertainty into a single decision region. This linking enables us to leverage existing methods in Bayesian active learning for robotic motion planning.

Chen et al. [5] provide a method to solve this problem by maximizing an objective function that satisfies *adaptive submodularity* [15] - a natural diminishing returns property that endows greedy policies with near-optimality guarantees. Unfortunately, naively applying this algorithm requires $\mathcal{O}\left(2^E\right)$ computation to select an edge to evaluate, where $E$ is the number of edges in all paths.

We define the Bern-DRD problem, which leverages additional structure in robotic motion planning by assuming edges are independent Bernoulli random variables [1], and regions correspond to sets of edges evaluating to true. We propose *Bernoulli Subregion Edge Cutting (*BiSECt*)*, which provides a greedy policy to select candidate edges in $\mathcal{O}\left(E\right)$. We prove our surrogate objective also satisfies adaptive submodularity [15], and provides the same bounds as Chen et al. [5] while being more efficient to compute.

We make the following contributions:

1. We show a novel equivalence between feasible path identification and the DRD problem, linking motion planning to Bayesian active learning.

2. We develop BiSECt, a near-optimal algorithm for the special case of Bernoulli tests, which selects tests in $\mathcal{O}\left(E\right)$ instead of $\mathcal{O}\left(2^E\right)$.

3. We demonstrate the efficacy of our algorithm on a spectrum of planning problems for mobile robots, manipulators, and real flight data collected from a full scale helicopter.

## 2 Problem Formulation

### 2.1 Planning as Feasible Path Identification on Explicit Graphs

Let $G = (V, E)$ be an explicit graph that consists of a set of vertices $V$ and edges $E$. Given a pair of start and goal vertices, $(v_s, v_g) \in V$, a search algorithm computes a path $\xi \subseteq E$ - a connected sequence of valid edges. To ascertain the validity of an edge, it invokes an evaluation function $\texttt{Eval} : E \to \{0, 1\}$. We address applications where edge evaluation is expensive, i.e., the computational cost $c(e)$ of computing $\texttt{Eval}(e)$ is significantly higher than regular search operations[2].

We define a world as an outcome vector $\mathbf{o} \in \{0, 1\}^{|E|}$ which assigns to each edge a boolean validity when evaluated, i.e. $\texttt{Eval}(e) = \mathbf{o}(e)$. We assume that the outcome vector is sampled from an independent Bernoulli distribution $P(\mathbf{o})$, giving rise to a *Generalized Binomial Graph (GBG)* [13].

We make a second simplification to the problem - from that of search to that of identification. Instead of searching $G$ online for a path, we frame the problem as identifying a valid path from a library of 'good' candidate paths $\Xi = (\xi_1, \xi_2, \ldots, \xi_m)$. The candidate set of paths $\Xi$ is constructed offline, while being cognizant of $P(\mathbf{o})$, and can be verified to ensure that all paths have acceptable solution quality when valid. [3] Hence we care about completeness with respect to $\Xi$ instead of $G$.

We wish to design an adaptive edge selector $\texttt{Select}(\mathbf{o})$ which is a decision tree that operates on a world $\mathbf{o}$, selects an edge for evaluation and branches on its outcome. The total cost of edge evaluation is $c(\texttt{Select}(\mathbf{o}))$. Our objective is to minimize the cost required to find a valid path:

$$\min \ \mathbb{E}_{\mathbf{o} \in P(\mathbf{o})} \left[ c(\texttt{Select}(\mathbf{o})) \right] \ \text{s.t} \ \forall \mathbf{o}, \exists \xi \ : \ \prod_{e \in \xi} \mathbf{o}(e) = 1 \ , \ \xi \subseteq \texttt{Select}(\mathbf{o}) \tag{1}$$

### 2.2 Decision Region Determination with Independent Bernoulli Tests

We now define an equivalent problem - *decision region determination with independent Bernoulli tests (Bern-DRD)*. Define a set of tests $\mathcal{T} = \{1, \ldots, n\}$, where the outcome of each test is a Bernoulli random variable $X_t \in \{0, 1\}$, $P(X_t = x_t) = \theta_t^{x_t}(1 - \theta_t)^{1-x_t}$. We define a set of hypotheses $h \in \mathcal{H}$, where each is an outcome vector $h \in \{0, 1\}^{\mathcal{T}}$ mapping all tests $t \in \mathcal{T}$ to outcomes $h(t)$. We define a set of regions $\{\mathcal{R}_i\}_{i=1}^m$, each of which is a subset of tests $\mathcal{R} \subseteq \mathcal{T}$. A region is determined to be valid if all tests in that region evaluate to true, which has probability $P(\mathcal{R}) = \prod_{t \in \mathcal{R}} P(X_t = 1)$.

If a set of tests $\mathcal{A} \subseteq \mathcal{T}$ are performed, let the observed outcome vector be denoted by $\mathbf{x}_{\mathcal{A}} \in \{0, 1\}^{|\mathcal{A}|}$. Let the *version space* $\mathcal{H}(\mathbf{x}_{\mathcal{A}})$ be the set of hypotheses consistent with observation vector $\mathbf{x}_{\mathcal{A}}$, i.e. $\mathcal{H}(\mathbf{x}_{\mathcal{A}}) = \{h \in \mathcal{H} \mid \forall t \in \mathcal{A}, h(t) = \mathbf{x}_{\mathcal{A}}(t)\}$.

We define a policy $\pi$ as a mapping from observation vector $\mathbf{x}_{\mathcal{A}}$ to tests. A policy terminates when it shows that at least one region is valid, or all regions are invalid. Let $\mathbf{x}_{\mathcal{T}} \in \{0, 1\}^{\mathcal{T}}$ be the ground truth - the outcome vector for all tests. Denote the observation vector of a policy $\pi$ given ground truth $\mathbf{x}_{\mathcal{T}}$ as $\mathbf{x}_{\mathcal{A}}(\pi, \mathbf{x}_{\mathcal{T}})$. The expected cost of a policy $\pi$ is $c(\pi) = \mathbb{E}_{\mathbf{x}_{\mathcal{T}}} \left[ c(\mathbf{x}_{\mathcal{A}}(\pi, \mathbf{x}_{\mathcal{T}}) \right]$ where $c(\mathbf{x}_{\mathcal{A}})$ is the cost of all tests $t \in \mathcal{A}$. The objective is to compute a policy $\pi^*$ with minimum cost that ensures at least one region is valid, i.e.

$$\pi^* \in \arg\min_{\pi} c(\pi) \ \text{s.t} \ \forall \mathbf{x}_{\mathcal{T}}, \exists \mathcal{R}_d \ : \ P(\mathcal{R}_d \mid \mathbf{x}_{\mathcal{A}}(\pi, \mathbf{x}_{\mathcal{T}})) = 1 \tag{2}$$

Note that we can cast problem (1) to (2) by setting $E = \mathcal{T}$ and $\Xi = \{\mathcal{R}_i\}_{i=1}^m$. That is, driving uncertainty into a region is equivalent to identification of a valid path (Fig. 2). This casting enables us to leverage efficient algorithms with near-optimality guarantees for motion planning.

## 3 Related Work

The computational bottleneck in motion planning varies with problem domain and that has led to a plethora of planning techniques ([23]). When vertex expansions are a bottleneck, A* [17] is optimally efficient while techniques such as partial expansions [28] address graph searches with large branching factors. The problem class we examine, that of expensive edge evaluation, has inspired a variety of

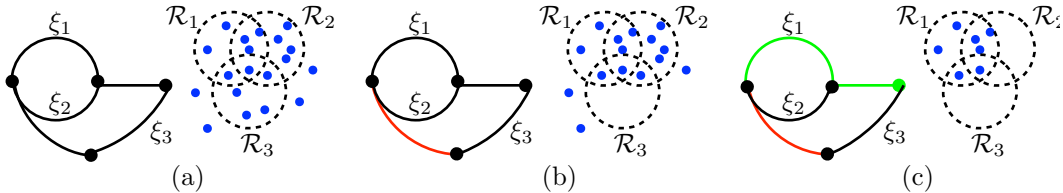

Figure 2: Equivalence between the feasible path identification problem and Bern-DRD. A path $\xi_i$ is equivalent to a region $\mathcal{R}_i$ over valid hypotheses (blue dots). Tests eliminate hypotheses and the algorithm terminates when uncertainty is pushed into a region ($\mathcal{R}_1$) and the corresponding path ($\xi_1$) is determined to be valid.

'lazy' approaches. The Lazy Probabilistic Roadmap (PRM) algorithm [1] only evaluates edges on the shortest path while Fuzzy PRM [26] evaluates paths that minimize probability of collision. The Lazy Weighted A* (LWA*) algorithm [8] delays edge evaluation in A* search and is reflected in similar techniques for randomized search [14, 6]. An approach most similar in style to ours is the LazyShortestPath (LazySP) framework [11] which examines the problem of which edges to evaluate on the shortest path. Instead of the finding the shortest path, our framework aims to efficiently identify a feasible path in a library of 'good' paths. Our framework is also similar to the Anytime Edge Evaluation (AEE*) framework [25] which deals with edge evaluation on a GBG. However, our framework terminates once a *single feasible path* is found while AEE* continues to evaluation in order to minimize expected cumulative sub-optimality bound. Similar to Choudhury et al. [7] and Burns and Brock [2], we leverage priors on the distribution of obstacles to make informed planning decisions.

We draw a novel connection between motion planning and optimal test selection which has a wide-spread application in medical diagnosis [21] and experiment design [4]. Optimizing the ideal metric, decision theoretic value of information [18], is known to be NP$^{\text{PP}}$ complete [22]. For hypothesis identification (known as the Optimal Decision Tree (ODT) problem), Generalized Binary Search (GBS) [10] provides a near-optimal policy. For disjoint region identification (known as the Equivalence Class Determination (ECD) problem), EC$^2$ [16] provides a near-optimal policy. When regions overlap (known as the Decision Region Determination (DRD) problem), HEC [19] provides a near-optimal policy. The DiRECT algorithm [5], a computationally more efficient alternative to HEC, forms the basis of our approach.

## 4 The Bernoulli Subregion Edge Cutting Algorithm

The DRD problem in general is addressed by the *Decision Region Edge Cutting (*DiRECT*)* [5] algorithm. The intuition behind the method is as follows - as tests are performed, hypotheses inconsistent with test outcomes are pruned away. Hence, tests should be incentivized to push the probability mass over hypotheses into *any region* as fast as possible. Chen et al. [5] derive a surrogate objective function that provides such an incentive by *creating separate sub-problems* for each region and combining them in a Noisy-OR fashion such that *quickly solving any one sub-problem suffices*. Importantly, this objective is adaptive submodular [15] - greedily maximizing such an objective results in a near-optimal policy.

We adapt the framework of DiRECT to address the Bern-DRD problem. We first provide a modification to the EC$^2$ sub-problem objective which is simpler to compute when the distribution over hypotheses is non-uniform, while providing the same guarantees. Unfortunately, naively applying DiRECT requires $\mathcal{O}\left(2^{\mathcal{T}}\right)$ computation per sub-problem. For the special case of independent Bernoulli tests, we present a more efficient *Bernoulli Subregion Edge Cutting (*BiSECT*)* algorithm, which computes each subproblem in $\mathcal{O}\left(\mathcal{T}\right)$ time. We provide a brief exposition deferring to the supplementary for detailed derivations.

### 4.1 A simple subproblem: One region versus all

Following Chen et al. [5], we define a 'one region versus all' subproblem, the solution of which helps address the Bern-DRD. Given a *single region*, the objective is to either push the version space to that region, or collapse it to a single hypothesis. We view a region $\mathcal{R}$ as a version space $\mathcal{R}^{\mathcal{H}} \subseteq \mathcal{H}$

consistent with its constituent tests. We define this subproblem over a set of *disjoint subregions* $\mathcal{S}_i$. Let the hypotheses in the target region $\mathcal{R}^{\mathcal{H}}$ be $\mathcal{S}_1$. Every other hypothesis $h \in \overline{\mathcal{R}^{\mathcal{H}}}$ is defined as its own subregion $\mathcal{S}_i, i > 1$, where $\overline{\mathcal{R}^{\mathcal{H}}}$ is a set of hypothesis where a region is *not* valid. Determining which subregion is valid falls under the framework of *Equivalence Class Determination* (ECD), (a special case of the DRD problem) and can be solved efficiently by the $EC^2$ algorithm (Golovin et al. [16]). This objective defines a graph with nodes as subregions and edges between distinct subregions, where the weight of an edge is the product of probabilities of subregions. As tests are performed and outcomes are received, the version space shrinks, and probabilities of different subregions are driven to 0. This has the effect of decreasing the total weight of edges. Importantly, the problem is solved i.f.f. the weight of all edges is zero. The weight over the set of subregions is:

$$w_{[16]}(\{\mathcal{S}_i\}) = \sum_{j \neq k} P(\mathcal{S}_j)P(\mathcal{S}_k) \tag{3}$$

When hypotheses have uniform weight, this can be computed efficiently for the 'one region versus all' subproblem. Let $P(\overline{\mathcal{S}_1}) = \sum_{i>1} P(\mathcal{S}_i)$:

$$w_{[16]}(\{\mathcal{S}_i\}) = P(\mathcal{S}_1)P(\overline{\mathcal{S}_1}) + P(\overline{\mathcal{S}_1})\left(P(\overline{\mathcal{S}_1}) - \frac{1}{|\mathcal{H}|}\right) \tag{4}$$

For non-uniform prior however, this quantity is more difficult to compute. We modify this objective slightly, adding self-edges on subregions $\mathcal{S}_i, i > 1$, enabling more efficient computation while still maintaining the same guarantees:

$$w_{\text{EC}}(\{\mathcal{S}_i\}) = P(\mathcal{S}_1)(\sum_{i \neq 1} P(\mathcal{S}_i)) + (\sum_{i \neq 1} P(\mathcal{S}_i))(\sum_{j \geq 1} P(\mathcal{S}_j))$$
$$= P(\mathcal{S}_1)P(\overline{\mathcal{S}_1}) + P(\overline{\mathcal{S}_1})^2 = P(\overline{\mathcal{R}^{\mathcal{H}}})(P(\mathcal{R}^{\mathcal{H}}) + P(\overline{\mathcal{R}^{\mathcal{H}}})) \tag{5}$$

For region $\mathcal{R}$, let the *relevant version space* be $\mathcal{H}^{\mathcal{R}}(\mathbf{x}_{\mathcal{A}}) = \{h \in \mathcal{H} \mid \forall t \in \mathcal{A} \cap \mathcal{R}, h(t) = \mathbf{x}_{\mathcal{A}}(t)\}$. The set of all hypotheses in $\mathcal{R}^{\mathcal{H}}$ consistent with relevant outcomes in $\mathbf{x}_{\mathcal{A}}$ is given by $\mathcal{R}^{\mathcal{H}} \cap \mathcal{H}^{\mathcal{R}}(\mathbf{x}_{\mathcal{A}})$. The terms $P(\mathcal{R}^{\mathcal{H}} \cap \mathcal{H}^{\mathcal{R}}(\mathbf{x}_{\mathcal{A}}))$ and $P(\overline{\mathcal{R}^{\mathcal{H}}} \cap \mathcal{H}^{\mathcal{R}}(\mathbf{x}_{\mathcal{A}}))$ allows us to quantify the progress made on determining region validity. Naively computing these terms would require computing all hypotheses and assigning them to correct subregions, thus requiring a runtime of $\mathcal{O}\left(2^{\mathcal{T}}\right)$. However, for the special case of Bernoulli tests, we can reduce this to $\mathcal{O}\left(\mathcal{T}\right)$ as we can see from the expression

$$w_{\text{EC}}(\{\mathcal{S}_i\} \cap \mathcal{H}^{\mathcal{R}}(\mathbf{x}_{\mathcal{A}})) = \left(1 - \prod_{i \in (\mathcal{R} \cap \mathcal{A})} \mathbb{I}(X_i = 1) \prod_{j \in (\mathcal{R} \setminus \mathcal{A})} \theta_j\right)\left(\prod_{k \in \mathcal{R} \cap \mathcal{A}} \theta_k^{\mathbf{x}_{\mathcal{A}}(k)}(1 - \theta_k)^{1 - \mathbf{x}_{\mathcal{A}}(k)}\right)^2 \tag{6}$$

We can further reduce this to $\mathcal{O}\left(1\right)$ when iteratively updated (see supplementary for derivations). We now define a criterion that incentivizes removing edges quickly and has theoretical guarantees. Let $f_{\text{EC}}(\mathbf{x}_{\mathcal{A}})$ be the weight of edges removed on observing outcome vector $\mathbf{x}_{\mathcal{A}}$. This is evaluated as

$$f_{\text{EC}}(\mathbf{x}_{\mathcal{A}}) = 1 - \frac{w_{\text{EC}}(\{\mathcal{S}_i\} \cap \mathcal{H}^{\mathcal{R}}(\mathbf{x}_{\mathcal{A}}))}{w_{\text{EC}}(\{\mathcal{S}_i\})}$$
$$= 1 - \frac{\left(1 - \prod_{i \in (\mathcal{R} \cap \mathcal{A})} \mathbb{I}(X_i = 1) \prod_{j \in (\mathcal{R} \setminus \mathcal{A})} \theta_j\right)\left(\prod_{k \in \mathcal{R} \cap \mathcal{A}} \theta_k^{\mathbf{x}_{\mathcal{A}}(k)}(1 - \theta_k)^{1 - \mathbf{x}_{\mathcal{A}}(k)}\right)^2}{1 - \prod_{i \in \mathcal{R}} \theta_i} \tag{7}$$

**Lemma 1.** *The expression $f_{\text{EC}}(\mathbf{x}_{\mathcal{A}})$ is strongly adaptive monotone and adaptive submodular.*

### 4.2 Solving the Bern-DRD problem using BISECT

We now return to Bern-DRD problem (2) where we have multiple regions $\{\mathcal{R}_1, \ldots, \mathcal{R}_m\}$ that overlap. Each region $\mathcal{R}_r$ is associated with an objective $f_{\text{EC}}^r(\mathbf{x}_{\mathcal{A}})$ for solving the 'one region versus all' problem. Since solving any one such subproblem suffices, we combine them in a *Noisy-OR*

**Algorithm 1:** Decision Region Determination with Independent Bernoulli Test($\{\mathcal{R}_i\}_{i=1}^m, \boldsymbol{\theta}, \mathbf{x}_\mathcal{T}$)

```
1  A ← ∅ ;
2  while (∄Rᵢ, P(Rᵢ|x_A) = 1) and (∃Rᵢ, P(Rᵢ|x_A) > 0) do
3  │   T_cand ← SelectCandTestSet(x_A) ;                  ▷ Using either (10) or (12)
4  │   t* ← SelectTest(T_cand, θ, x_A) ;      ▷ Using either (11),(13),(14),(15) or (16)
5  │   A ← A ∪ t* ;
6  │   x_{t*} ← x_T(t*) ;                                 ▷ Observe outcome for selected test
```

formulation by defining an objective $f_{\mathrm{DRD}}(\mathbf{x}_\mathcal{A}) = 1 - \prod_{r=1}^m (1 - f_{\mathrm{EC}}^r(\mathbf{x}_\mathcal{A}))$ [5] which evaluates to

$$1 - \prod_{r=1}^m \left( \frac{\left(1 - \prod_{i \in (\mathcal{R}_r \cap \mathcal{A})} \mathbb{I}(X_i = 1) \prod_{j \in (\mathcal{R}_r \setminus \mathcal{A})} \theta_j \right)\left(\prod_{k \in \mathcal{R}_r \cap \mathcal{A}} \theta_k^{\mathbf{x}_\mathcal{A}(k)}(1 - \theta_k)^{1 - \mathbf{x}_\mathcal{A}(k)}\right)^2}{1 - \prod_{i \in \mathcal{R}_r} \theta_i} \right) \quad (8)$$

Since $f_{\mathrm{DRD}}(\mathbf{x}_\mathcal{A}) = 1$ iff $f_{\mathrm{EC}}^r(\mathbf{x}_\mathcal{A}) = 1$ for at least one $r$, we define the following surrogate problem to Bern-DRD

$$\pi^* \in \arg\min_\pi c(\pi) \text{ s.t } \forall \mathbf{x}_\mathcal{T} \; : \; f_{\mathrm{DRD}}(\mathbf{x}_\mathcal{A}(\pi, \mathbf{x}_\mathcal{T})) \geq 1 \quad (9)$$

The surrogate problem has a structure that allows greedy policies to have near-optimality guarantees
**Lemma 2.** *The expression $f_{\mathrm{DRD}}(\mathbf{x}_\mathcal{A})$ is strongly adaptive monotone and adaptive submodular.*
**Theorem 1.** *Let $m$ be the number of regions, $p_{\min}^h$ the minimum prior probability of any hypothesis, $\pi_{DRD}$ be the greedy policy and $\pi^*$ with the optimal policy. Then $c(\pi_{DRD}) \leq c(\pi^*)(2m\log\frac{1}{p_{\min}^h} + 1)$.*

We now describe the BISECT algorithm. Algorithm 1 shows the framework for a general decision region determination algorithm. In order to specify BISECT, we need to define two options - a candidate test set selection function $\mathtt{SelectCandTestSet}(\mathbf{x}_\mathcal{A})$ and a test selection function $\mathtt{SelectTest}(\mathcal{T}_{\mathrm{cand}}, \boldsymbol{\theta}, \mathbf{x}_\mathcal{A})$. The unconstrained version of BISECT implements $\mathtt{SelectCandTestSet}(\mathbf{x}_\mathcal{A})$ to return the set of all tests $\mathcal{T}_{\mathrm{cand}}$ that contains only unevaluated tests belonging to active regions

$$\mathcal{T}_{\mathrm{cand}} = \left\{ \bigcup_{i=1}^m \{\mathcal{R}_i \mid P(\mathcal{R}_i|\mathbf{x}_\mathcal{A}) > 0\} \right\} \setminus \mathcal{A} \quad (10)$$

We now examine the BISECT test selection rule $\mathtt{SelectTest}(\mathcal{T}_{\mathrm{cand}}, \boldsymbol{\theta}, \mathbf{x}_\mathcal{A})$

$$
\begin{aligned}
t^* \in \arg\max_{t \in \mathcal{T}_{\mathrm{cand}}} \frac{1}{c(t)} \mathbb{E}_{x_t} &\left[ \prod_{r=1}^m \left( 1 - \prod_{i \in (\mathcal{R}_r \cap \mathcal{A})} \mathbb{I}(X_i = 1) \prod_{j \in (\mathcal{R}_r \setminus \mathcal{A})} \theta_j \right) \right. \\
&\left. - \left( \prod_{r=1}^m \left( 1 - \prod_{i \in (\mathcal{R}_r \cap \mathcal{A} \cup t)} \mathbb{I}(X_i = 1) \prod_{j \in (\mathcal{R}_r \setminus \mathcal{A} \cup t)} \theta_j \right) \right) \left( \theta_t^{x_t}(1 - \theta_t)^{1 - x_t} \right)^{2 \sum_{k=1}^m \mathbb{I}(t \in \mathcal{R}_k)} \right]
\end{aligned} \quad (11)
$$

The intuition behind this update is that tests are selected to squash the probability of regions not being valid. It also additionally incentivizes selection of tests on which multiple regions overlap.

## 4.3 Adaptively constraining test selection to most likely region

We observe in our experiments that the surrogate (8) suffers from a slow convergence problem - $f_{\mathrm{DRD}}(\mathbf{x}_\mathcal{A})$ takes a long time to converge to 1 when greedily optimized. To alleviate the convergence problem, we introduce an alternate candidate selection function $\mathtt{SelectCandTestSet}(\mathbf{x}_\mathcal{A})$ that assigns to $\mathcal{T}_{\mathrm{cand}}$ the set of all tests that belong to the most likely region $\mathcal{T}_{\mathrm{maxP}}$ which is evaluated as follows (we will refer to this variant as MAXPROBREG)

$$\mathcal{T}_{\mathrm{maxP}} = \left\{ \arg\max_{\mathcal{R}_i = (\mathcal{R}_1, \mathcal{R}_2, \ldots, \mathcal{R}_m)} P(\mathcal{R}_i|\mathbf{x}_\mathcal{A}) \right\} \setminus \mathcal{A} \quad (12)$$

Applying the constraint in (12) leads to a dramatic improvement for any test selection policy as we will show in Sec. 5.2. The following theorem offers a partial explanation

**Theorem 2.** *A policy that greedily latches to a region according the the posterior conditioned on the region outcomes has a near-optimality guarantee of 4 w.r.t the optimal region evaluation sequence.*

Applying the constraint in (12) implies we are no longer greedily optimizing $f_{\text{DRD}}(\mathbf{x}_\mathcal{A})$. However, the following theorem bounds the sub-optimality of this policy.

**Theorem 3.** *Let $p_{\min} = \min_i P(\mathcal{R}_i)$, $p_{\min}^h = \min_{h \in \mathcal{H}} P(h)$ and $l = \max_i |\mathcal{R}_i|$. The policy using (12) has a suboptimality of $\alpha \left( 2m \log \left( \frac{1}{p_{\min}^h} \right) + 1 \right)$ where $\alpha \leq \left( 1 - \max \left( (1 - p_{\min})^2, p_{\min}^{\frac{2}{l}} \right) \right)^{-1}$.*

## 5   Experiments

We evaluate BISECT on a collection of datasets spanning across a spectrum of synthetic problems and real-world planning applications. The synthetic problems are created by randomly selecting problem parameters to test the general applicability of BISECT. The motion planning datasets range from simplistic yet insightful 2D problems to more realistic high dimension problems as encountered by an UAV or a robot arm. The 7D arm planning dataset is obtained from a high fidelity simulation as shown in Fig. 4(a). Finally, we test BISECT on experimental data collected from a full scale helicopter flying that has to avoid unmapped wires at high speed as it comes into land as shown in Fig. 4(b). Refer to supplementary for exhaustive details on experiments and additional results. Open-source code and details can be found here: `https://github.com/sanjibac/matlab_learning_collision_checking`

### 5.1   Heuristic approaches to solving the Bern-DRD problem

We propose a collection of competitive heuristics that can also be used to solve the Bern-DRD problem. These heuristics are various `SelectTest`$(\mathcal{T}_{\text{cand}}, \theta, \mathbf{x}_\mathcal{A})$ policies in the framework of Alg. 1. To simplify the setting, we assume unit cost $c(t) = 1$ although it would be possible to extend these to nonuniform setting. The first heuristic RANDOM selects a test by sampling uniform randomly

$$t^* \in \mathcal{T}_{\text{cand}} \tag{13}$$

We adopt our next heuristic MAXTALLY from Dellin and Srinivasa [11] where the test belonging to most regions is selected. It uses the following criteria, which exhibits a 'fail-fast' characteristic

$$t^* \in \underset{t \in \mathcal{T}_{\text{cand}}}{\arg\max} \sum_{i=1}^m \mathbb{I}\left( t \in \mathcal{R}_i, P(\mathcal{R}_i | \mathbf{x}_\mathcal{A}) > 0 \right) \tag{14}$$

The next policy SETCOVER selects tests that maximize the expected number of 'covered' tests, i.e. if a selected test is in collision, how many other tests does it remove from consideration.

$$t^* \in \underset{t \in \mathcal{T}_{\text{cand}}}{\arg\max} \, (1 - \theta_t) \left| \left\{ \bigcup_{i=1}^m \{ \mathcal{R}_i \mid P(\mathcal{R}_i | \mathbf{x}_\mathcal{A}) > 0 \} - \bigcup_{j=1}^m \{ \mathcal{R}_j \mid P(\mathcal{R}_j |, \underset{X_t = 0}{\mathbf{x}_\mathcal{A}}) > 0 \} \right\} \setminus \{ \mathcal{A} \cup \{t\} \} \right| \tag{15}$$

**Theorem 4.** SETCOVER *is a near-optimal policy for the problem of optimally checking all regions.*

The last heuristic is derived from a classic heuristic in decision theory: myopic value of information (Howard [18]). MVOI greedily chooses the test that maximizes the change in the probability mass of the most likely region. This test selection works only with `SelectCandTestSet`$(\mathbf{x}_\mathcal{A}) = \mathcal{T}_{\text{maxP}}$.

$$t^* \in \underset{t \in \mathcal{T}_{\text{maxP}}}{\arg\max} \, (1 - \theta_t) \max_{i=1,\ldots,m} P(\mathcal{R}_i \mid \mathbf{x}_\mathcal{A}, X_t = 0) \tag{16}$$

We also evaluate against state of the art LAZYSP [11] planner which explicitly minimizes collision checking effort while trying to guarantee optimality. We ran two variants of LazySP. The first variant is the vanilla unconstrained algorithm that searches for the shortest path on the entire graph, collision checks the path and repeats. The second variant is constrained to the library of paths used by all other baselines.

### 5.2   Analysis of results

Table 1 shows the evaluation cost of all algorithms on various datasets normalized w.r.t BISECT. The two numbers are lower and upper $95\%$ confidence intervals - hence it conveys how much fractionally poorer are algorithms w.r.t BISECT. The best performance on each dataset is highlighted. We present a set of observations to interpret these results.

**O 1.** BISECT *has a consistently competitive performance across all datasets.*

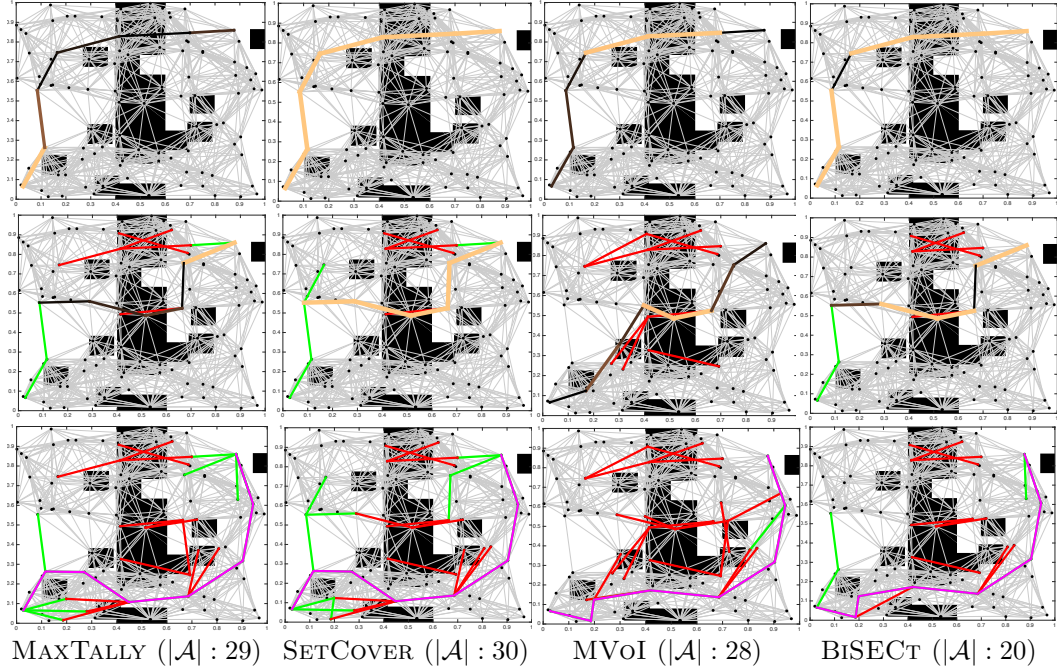

MAXTALLY ($|\mathcal{A}|$ : 29)    SETCOVER ($|\mathcal{A}|$ : 30)    MVOI ($|\mathcal{A}|$ : 28)    BISECT ($|\mathcal{A}|$ : 20)

Figure 3: Performance (number of evaluated edges) of all algorithms on 2D geometric planning. Snapshots, at start, interim and final stages respectively, show evaluated valid edges (green), invalid edges (red) and the final path (magenta). The utility of edges as computed by algorithms is shown varying from low (black) to high (cream).

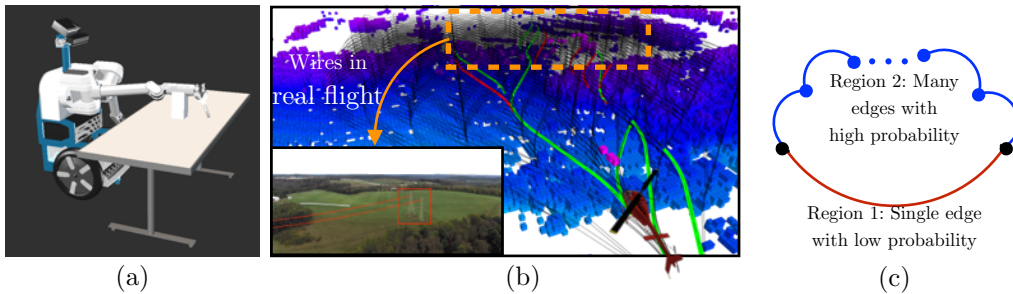

(a)    (b)    (c)

Figure 4: (a) A 7D arm has to perform pick and place tasks at high speed in a table with clutter. (b) Experimental data from a full-scale helicopter that has to react quickly to avoid unmapped wires detected by the sensor. BISECT (given an informative prior) checks a small number of edges around the detected wire and identifies a path. (c) Scenario where regions have size disparity. Unconstrained BISECT significantly outperforms other algorithms on such a scenario.

Table 1 shows that on 13 out of the 14 datasets, BISECT is at par with the best. On 7 of those it is exclusively the best.

**O 2.** *The* MAXPROBREG *variant improves the performance of all algorithms on most datasets*

Table 1 shows that this is true on 12 datasets. The impact is greatest on RANDOM on the 2D Forest dataset - performance improves from $(19.45, 27.66)$ to $(0.13, 0.30)$. However, this is not true in general. On datasets with large disparity in region sizes as illustrated in Fig. 4(c), unconstrained BISECT significantly outperforms other algorithms. In such scenarios, MAXPROBREG latches on to the most probable path which also happens to have a large number of edges. It performs poorly on instances where this region is invalid, while the other region containing a single edge is valid. Unconstrained BISECT prefers to evaluate the single edge belonging to region 1 before proceeding to evaluate region 2, performing optimally on those instances. Hence, the myopic nature of MAXPROBREG is the reason behind its poor performance.

**O 3.** *On planning problems,* BISECT *strikes a trade-off between the complimentary natures of* MAXTALLY *and* MVOI.

Table 1: Normalized evaluation cost - (lower, upper) bound of 95% confidence interval

| | LAZYSP | MVOI | RANDOM | MAXTALLY | SETCOVER | BISECT |
|---|---|---|---|---|---|---|
| | Unconstrained | | Unconstrained | Unconstrained | Unconstrained | Unconstrained |
| | Constrained | | MaxProbReg | MaxProbReg | MaxProbReg | MaxProbReg |
| **Synthetic Bernoulli Test: Variation across region overlap** | | | | | | |
| Small | | | (4.18, 6.67) | (3.49, 5.23) | (1.77, 3.01) | (1.42, 2.36) |
| $m:100$ | | (0.00, 0.08) | (0.12, 0.29) | (0.12, 0.25) | (0.18, 0.40) | (0.00, 0.00) |
| Medium | | | (3.27, 4.40) | (3.04, 4.30) | (3.55, 4.67) | (1.77, 2.64) |
| $m:500$ | | (0.00, 0.00) | (0.05, 0.25) | (0.14, 0.24) | (0.14, 0.33) | (0.00, 0.00) |
| Large | | | (2.86, 4.26) | (2.62, 3.85) | (2.94, 3.71) | (1.33, 1.81) |
| $m:1e3$ | | (−0.11, 0.00) | (0.00, 0.28) | (0.06, 0.26) | (0.09, 0.22) | (0.00, 0.00) |
| **2D Geometric Planning: Variation across environments** | | | | | | |
| Forest | (10.8, 14.3) | | (19.5, 27.7) | (4.68, 6.55) | (3.53, 5.07) | (1.90, 2.46) |
| | (1.38, 2.51) | (0.03, 0.18) | (0.13, 0.30) | (0.09, 0.18) | (0.00, 0.09) | (0.00, 0.00) |
| OneWall | (6.96, 11.3) | | (13.4, 17.8) | (4.12, 4.89) | (1.36, 2.11) | (0.76, 1.20) |
| | (0.16, 0.55) | (0.045, 0.21) | (0.11, 0.42) | (0.00, 0.12) | (0.14, 0.29) | (0.00, 0.00) |
| TwoWall | (18.9, 25.6) | | (13.8, 16.6) | (2.76, 3.93) | (2.07, 2.94) | (0.91, 1.44) |
| | (−0.17, 0.01) | (0.00, 0.09) | (0.33, 0.51) | (0.10, 0.20) | (0.00, 0.00) | (0.00, 0.00) |
| **2D Geometric Planning: Variation across region size** | | | | | | |
| OneWall | (5.82, 12.1) | | (12.1, 16.0) | (4.47, 5.13) | (2.00, 3.41) | (0.94, 1.42) |
| $m:300$ | (0.00, 0.57) | (0.00, 0.17) | (0.12, 0.42) | (0.06, 0.24) | (0.00, 0.38) | (0.00, 0.00) |
| OneWall | (5.43, 10.02) | | (13.3, 16.8) | (2.18, 3.77) | (1.04, 1.62) | (0.41, 0.91) |
| $m:858$ | (−0.03, 0.45) | (0.00, 0.14) | (0.09, 0.27) | (−0.04, 0.08) | (0.00, 0.14) | (0.00, 0.00) |
| **Non-holonomic Path Planning: Variation across environments** | | | | | | |
| Forest | (1.97, 3.81) | | (22.4, 29.7) | (9.79, 11.14) | (2.63, 5.28) | (1.54, 2.46) |
| | (0.15, 0.47) | (0.09, 0.18) | (0.46, 0.79) | (0.25, 0.38) | (0.00, 0.00) | (0.00, 0.00) |
| OneWall | (0.97, 2.45) | | (13.0, 15.8) | (8.40, 11.47) | (3.72, 4.54) | (3.28, 3.78) |
| | (0.02, 0.51) | (−0.11, 0.11) | (0.00, 0.12) | (0.21, 0.28) | (−0.11, 0.11) | (0.00, 0.00) |
| **7D Arm Planning: Variation across environments** | | | | | | |
| Table | (0.97, 1.59) | | (15.1, 19.4) | (4.80, 6.98) | (1.36, 2.17) | (0.32, 0.67) |
| | (0.24, 0.72) | (0.28, 0.54) | (0.13, 0.31) | (0.00, 0.04) | (0.00, 0.11) | (0.00, 0.00) |
| Clutter | (0.28, 1.19) | | (7.92, 9.85) | (3.96, 6.44) | (1.42, 2.07) | (1.23, 1.75) |
| | (0.00, 0.38) | (0.02, 0.20) | (0.14, 0.36) | (0.00, 0.00) | (0.00, 0.11) | (0.00, 0.00) |
| **Datasets with large disparity in region sizes** | | | | | | |
| Synth. | | | (6.50, 8.00) | (5.50, 6.50) | (3.00, 3.50) | (0.00, 0.00) |
| $(T:10)$ | | (3.00, 3.50) | (3.00, 4.50) | (5.00, 7.50) | (3.00, 3.50) | (3.00, 3.50) |
| 2D Plan | | | (9.50, 11.3) | (2.80, 6.10) | (6.60, 10.5) | (0.00, 0.00) |
| $(m:2)$ | | (6.60, 10.5) | (6.90, 10.8) | (6.80, 8.30) | (6.60, 10.5) | (7.30, 11.2) |

We examine this in the context of 2D planning as shown in Fig. 3. MAXTALLY selects edges belonging to many paths which is useful for path elimination but does not reason about the event when the edge is not in collision. MVOI selects edges to eliminate the most probable path but does not reason about how many paths a single edge can eliminate. BISECT switches between these behaviors thus achieving greater efficiency than both heuristics. **O 4.** BISECT *checks informative edges in collision avoidance problems encountered a helicopter*

Fig. 4(b) shows the efficacy of BISECT on experimental flight data from a helicopter avoiding wire.

# 6 Conclusion

In this paper, we addressed the problem of identification of a feasible path from a library while minimizing the expected cost of edge evaluation given priors on the likelihood of edge validity. We showed that this problem is equivalent to a decision region determination problem where the goal is to select tests (edges) that drive uncertainty into a single decision region (a valid path). We proposed BISECT, and efficient and near-optimal algorithm that solves this problem by greedily optimizing a surrogate objective. We validated BISECT on a spectrum of problems against state of the art heuristics and showed that it has a consistent performance across datasets. This works serves as a first step towards importing Bayesian active learning approaches into the domain of motion planning.

**Acknowledgments**

We would like to acknowledge the support from ONR grant N000141310821. We would like to thank Shushman Choudhury for insightful discussions and the 7D arm planning datasets. We would like to thank Oren Salzaman, Mohak Bhardwaj, Vishal Dugar and Paloma Sodhi for feedback on the paper.

## Footnotes

[1] Generally, edges in this graph are correlated, as edges in collision are likely to have neighbours in collision. Unfortunately, even measuring this correlation is challenging, especially in the high-dimensional non-linear configuration space of robot arms. Assuming independent edges is a common simplification [23, 25, 7, 2, 11]

[2]It is assumed that $c(e)$ is modular and non-zero. It can scale with edge length.

[3]Refer to supplementary on various methods to construct a library of good candidate paths

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
