[Supplementary Material · supplementary.pdf]

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

## 4.1 Preliminaries: Hypothesis as outcome vectors

In order to apply the DRD framework of Chen et al. [5], we need to view regions as a sets of hypotheses. A hypothesis $h$ is a mapping from a test $t \in \mathcal{T}$ to an outcome $h(t)$ and is defined as an outcome vector $h \in \{0,1\}^{\mathcal{T}}$. We use the symbol $\mathcal{H}$ to denote the set of all hypothesis ($\mathcal{H} = \{0,1\}^{\mathcal{T}}$). Using the independent Bernoulli distribution, the probability of a hypothesis is $P(h) = \prod_{t \in \mathcal{T}} P(X_t = h(t)) = \prod_{t \in \mathcal{T}} \theta_t^{h(t)}(1 - \theta_t)^{1-h(t)}$.

Given a observation vector $\mathbf{x}_{\mathcal{A}}$, let the *version space* $\mathcal{H}(\mathbf{x}_{\mathcal{A}})$ be the set of hypothesis consistent with $\mathbf{x}_{\mathcal{A}}$, i.e. $\mathcal{H}(\mathbf{x}_{\mathcal{A}}) = \{h \in \mathcal{H} \mid \forall t \in \mathcal{A}, h(t) = \mathbf{x}_{\mathcal{A}}(t)\}$. The probability mass of all the version space can evaluated as $P(\mathcal{H}(\mathbf{x}_{\mathcal{A}})) = \sum_{h \in \mathcal{H}(\mathbf{x}_{\mathcal{A}})} P(h) = \prod_{i \in \mathcal{A}} \theta_i^{\mathbf{x}_{\mathcal{A}}(i)}(1 - \theta_i)^{1-\mathbf{x}_{\mathcal{A}}(i)}$

Although we initially defined a region as a clause on constituent test outcomes being true, we can now view them as a version space consistent with the constituent tests. Hence given a region $\mathcal{R}$, we define the version space $\mathcal{R}^{\mathcal{H}} \in \mathcal{H}$ as a set of consistent hypothesis $\mathcal{R}^{\mathcal{H}} = \{h \in \mathcal{H} \mid \forall t \in \mathcal{R}, h(t) = 1\}$

Hence the probability of a region being valid is the probability mass of all consistent hypothesis
$$P(\mathcal{R}^{\mathcal{H}}) = \sum_{h \in \mathcal{R}^{\mathcal{H}}} P(h) = \prod_{i \in \mathcal{R}} P(X_i = 1) = \prod_{i \in \mathcal{R}} \theta_i$$

We will now define a set of useful expressions that will be used by BISECT. Given a observation vector $\mathbf{x}_{\mathcal{A}}$, the *relevant version space* is denoted as $\mathcal{H}^{\mathcal{R}}(\mathbf{x}_{\mathcal{A}}) = \{h \in \mathcal{H} \mid \forall t \in \mathcal{A} \cap \mathcal{R}, h(t) = \mathbf{x}_{\mathcal{A}}(t)\}$. Hence the set of all hypothesis in $\mathcal{R}^{\mathcal{H}}$ consistent with relevant outcomes in $\mathbf{x}_{\mathcal{A}}$ is given by $\mathcal{R}^{\mathcal{H}} \cap \mathcal{H}^{\mathcal{R}}(\mathbf{x}_{\mathcal{A}})$. The probability $P(\mathcal{R}^{\mathcal{H}} \cap \mathcal{H}^{\mathcal{R}}(\mathbf{x}_{\mathcal{A}}))$ is as follows

$$
\begin{aligned}
P(\mathcal{R}^{\mathcal{H}} \cap \mathcal{H}^{\mathcal{R}}(\mathbf{x}_{\mathcal{A}})) &= \sum_{h \in \mathcal{R}^{\mathcal{H}} \cap \mathcal{H}^{\mathcal{R}}(\mathbf{x}_{\mathcal{A}})} P(h) \\
&= \sum_{h \in \mathcal{R}^{\mathcal{H}} \cap \mathcal{H}^{\mathcal{R}}(\mathbf{x}_{\mathcal{A}})} \prod_{i \in \mathcal{T}} P(X_i = h(i)) \\
&= \prod_{i \in (\mathcal{R} \cap \mathcal{A})} \mathbb{I}(X_i = 1) \prod_{j \in (\mathcal{R} \backslash \mathcal{A})} P(X_j = 1) \prod_{k \in \mathcal{R} \cap \mathcal{A}} P(X_k = \mathbf{x}_{\mathcal{A}}(k)) \\
&= \prod_{i \in (\mathcal{R} \cap \mathcal{A})} \mathbb{I}(X_i = 1) \prod_{j \in (\mathcal{R} \backslash \mathcal{A})} \theta_j \prod_{k \in \mathcal{R} \cap \mathcal{A}} \theta_k^{\mathbf{x}_{\mathcal{A}}(k)} (1 - \theta_k)^{1 - \mathbf{x}_{\mathcal{A}}(k)}
\end{aligned}
\tag{3}
$$

We will now derive similar expressions for the probability of a region *not* being valid. The probability mass of hypothesis where a region $\mathcal{R}$ is not valid is $P(\overline{\mathcal{R}^{\mathcal{H}}}) = \sum_{h \in \overline{\mathcal{R}^{\mathcal{H}}}} P(h) = 1 - \prod_{i \in \mathcal{R}} \theta_i$

Similarly, the set of all hypothesis in $\overline{\mathcal{R}^{\mathcal{H}}}$ consistent with relevant outcomes in $\mathbf{x}_{\mathcal{A}}$ is given by $\overline{\mathcal{R}^{\mathcal{H}}} \cap \mathcal{H}^{\mathcal{R}}(\mathbf{x}_{\mathcal{A}})$. The probability $P(\overline{\mathcal{R}^{\mathcal{H}}} \cap \mathcal{H}^{\mathcal{R}}(\mathbf{x}_{\mathcal{A}}))$ is as follows

$$
\begin{aligned}
P(\overline{\mathcal{R}^{\mathcal{H}}} \cap \mathcal{H}^{\mathcal{R}}(\mathbf{x}_{\mathcal{A}})) &= \sum_{h \in \overline{\mathcal{R}^{\mathcal{H}}} \cap \mathcal{H}^{\mathcal{R}}(\mathbf{x}_{\mathcal{A}})} P(h) \\
&= \sum_{h \in \overline{\mathcal{R}^{\mathcal{H}}} \cap \mathcal{H}^{\mathcal{R}}(\mathbf{x}_{\mathcal{A}})} \prod_{i \in \mathcal{T}} P(X_i = h(i)) \\
&= \left( 1 - \prod_{i \in (\mathcal{R} \cap \mathcal{A})} \mathbb{I}(X_i = 1) \prod_{j \in (\mathcal{R} \backslash \mathcal{A})} P(X_j = 1) \right) \prod_{k \in \mathcal{R} \cap \mathcal{A}} P(X_k = \mathbf{x}_{\mathcal{A}}(k)) \\
&= \left( 1 - \prod_{i \in (\mathcal{R} \cap \mathcal{A})} \mathbb{I}(X_i = 1) \prod_{j \in (\mathcal{R} \backslash \mathcal{A})} \theta_j \right) \prod_{k \in \mathcal{R} \cap \mathcal{A}} \theta_k^{\mathbf{x}_{\mathcal{A}}(k)} (1 - \theta_k)^{1 - \mathbf{x}_{\mathcal{A}}(k)}
\end{aligned}
\tag{4}
$$

### 4.2 A simple subproblem: One region versus all

We will now define a simple subproblem whose solution will help in addressing the Bern-DRD problem. We define the 'one region versus all' subproblem as follows - given a *single region*, the objective is to either push the entire probability mass of the version space on a region or collapse it on a single relevant hypothesis. We will view this as a decision problem on the space of *disjoint subregions*.

We refer to hypothesis region $\mathcal{R}^{\mathcal{H}}$ as subregion $\mathcal{S}_1$ as shown in Fig.3. Every other hypothesis $h \in \overline{\mathcal{R}^{\mathcal{H}}}$ is defined as its own subregion $\mathcal{S}_i$. Determining which subregion is valid falls under the framework of *Equivalence Class Determination* (ECD), (a special case of the DRD problem) and can be solved efficiently by the EC$^2$ algorithm (Golovin et al. [17]).

#### 4.2.1 The EC$^2$ algorithm

The ECD problem is a special case of the DRD problem described in (2) to a case where regions are disjoint. In order to avoid confusion with DRD regions, we will hence forth refer to them as sub-regions. Let $\{\mathcal{S}_1, \ldots, \mathcal{S}_l\}$ be a set of disjoint subregions, i.e, $\mathcal{S}_i \cap \mathcal{S}_j = 0$ for $i \neq j$. Golovin et al.

[17] provide an efficient yet near-optimal criterion for solving ECD in their EC$^2$ algorithm which we discuss in brief.

The EC$^2$ algorithm defines a graph $\mathcal{G} = (\mathcal{V}, \mathcal{E})$ where the nodes are hypotheses and edges are between hypotheses in different decision regions $E = \cup_{i \neq j} \{\{h, h'\} \mid h \in \mathcal{S}_i, h' \in \mathcal{S}_j\}$. The weight of an edge is defined as $w(\{h, h'\}) = P(h)P(h')$. The weight of a set of edges is defined as $w(\mathcal{E}') = \sum_{\varepsilon \in \mathcal{E}'} w(\varepsilon)$. An edge is said to be 'cut' by an observation if either hypothesis is inconsistent with the observation. Hence a test $t$ with outcome $x_t$ is said to cut a set of edges $\mathcal{E}(x_t) = \{\{h, h'\} \mid h(t) \neq x_t \vee h'(t) \neq x_t\}$. The aim is to cut all edges by performing test while minimizing cost. Before we describe the objective, we first specify how EC$^2$ efficiently computes weights by defininig a weight function over subregions.

$$w_{[17]}(\{\mathcal{S}_i\}) = \sum_{i \neq j} P(\mathcal{S}_i)P(\mathcal{S}_j) \tag{5}$$

When hypotheses have uniform weight, this can be computed efficiently for the 'one region versus all' subproblem. Let $P(\overline{\mathcal{S}_1}) = \sum_{i > 1} P(\mathcal{S}_i)$:

$$w_{[17]}(\{\mathcal{S}_i\}) = P(\mathcal{S}_1)P(\overline{\mathcal{S}_1}) + P(\overline{\mathcal{S}_1})\left(P(\overline{\mathcal{S}_1}) - \frac{1}{|\mathcal{H}|}\right) \tag{6}$$

EC$^2$ defines an objective function $f_{\mathrm{EC}}(\mathbf{x}_\mathcal{A})$ that measures the weight of edges cut. This is the difference between the original weight of subregions $\mathcal{S}_i$ and the weight of pruned subregions $\mathcal{S}_i \cap \mathcal{H}(\mathbf{x}_\mathcal{A})$, i.e. $f_{\mathrm{EC}}(\mathbf{x}_\mathcal{A}) = w_{[17]}(\{\mathcal{S}_i\}) - w_{[17]}(\{\mathcal{S}_i\} \cap \mathcal{H}(\mathbf{x}_\mathcal{A}))$.

EC$^2$ uses the fact that $f_{\mathrm{EC}}(\mathbf{x}_\mathcal{A})$ is *adaptive submodular* (Golovin and Krause [16]) to define a greedy algorithm. Let the expected marginal gain of a test be $\Delta_{f_{\mathrm{EC}}}(t \mid x) = \mathbb{E}_{x_t}\left[f_{\mathrm{EC}}(\mathbf{x}_{\mathcal{A} \cup \{t\}}) - f_{\mathrm{EC}}(\mathbf{x}_\mathcal{A}) \mid \mathbf{x}_\mathcal{A}\right]$. EC$^2$ greedily selects a test $t^* \in \arg\max_t \frac{\Delta_{f_{\mathrm{EC}}}(t \mid \mathbf{x}_\mathcal{A})}{c(t)}$.

### 4.2.2 An alternative to EC$^2$ on the 'one region versus all' problem

Figure 3: The 'one region versus all' ECD problem. The region $\mathcal{R}^\mathcal{H}$ is shown as a circle encompassing a set of consistent hypothesis $h$ (green dots). Hypothesis for which the region is not valid lie outside the circle (dots in colors other than green). The objective is to compute an efficient policy to either force the probability mass in the region $\mathcal{R}^\mathcal{H}$ or determine the *unique* hypothesis $h \in \overline{\mathcal{R}^\mathcal{H}}$.

For non-uniform prior the quantity (6) is more difficult to compute. We modify this objective slightly, adding self-edges on subregions $\mathcal{S}_i, i > 1$ as shown in Fig. 3, enabling more efficient computation while still maintaining the same guarantees:

$$\begin{aligned} w_{\mathrm{EC}}(\{\mathcal{S}_i\}) &= P(\mathcal{S}_1)(\sum_{i \neq 1} P(\mathcal{S}_i)) + (\sum_{i \neq 1} P(\mathcal{S}_i))(\sum_{j \geq i} P(\mathcal{S}_j)) \\ &= P(\mathcal{S}_1)P(\overline{\mathcal{S}_1}) + P(\overline{\mathcal{S}_1})^2 \\ &= P(\mathcal{R}^\mathcal{H})P(\overline{\mathcal{R}^\mathcal{H}}) + P(\overline{\mathcal{R}^\mathcal{H}})P(\overline{\mathcal{R}^\mathcal{H}}) \\ &= P(\overline{\mathcal{R}^\mathcal{H}})(P(\mathcal{R}^\mathcal{H}) + P(\overline{\mathcal{R}^\mathcal{H}})) \\ &= 1 - \prod_{i \in \mathcal{R}} \theta_i \end{aligned} \tag{7}$$

Similarly we can compute $w_{\text{EC}}(\{\mathcal{S}_i\} \cap \mathcal{H}^{\mathcal{R}}(\mathbf{x}_\mathcal{A}))$ using (3) and (4)

$$
\begin{aligned}
& w_{\text{EC}}(\{\mathcal{S}_i\} \cap \mathcal{H}^{\mathcal{R}}(\mathbf{x}_\mathcal{A})) \\
&= P(\mathcal{S}_1 \cap \mathcal{H}^{\mathcal{R}}(\mathbf{x}_\mathcal{A}))P(\overline{\mathcal{S}_1} \cap \mathcal{H}^{\mathcal{R}}(\mathbf{x}_\mathcal{A})) + P(\overline{\mathcal{S}_1} \cap \mathcal{H}^{\mathcal{R}}(\mathbf{x}_\mathcal{A}))^2 \\
&= P(\mathcal{R} \cap \mathcal{H}^{\mathcal{R}}(\mathbf{x}_\mathcal{A}))P(\overline{\mathcal{R}} \cap \mathcal{H}^{\mathcal{R}}(\mathbf{x}_\mathcal{A})) + P(\overline{\mathcal{R}} \cap \mathcal{H}^{\mathcal{R}}(\mathbf{x}_\mathcal{A}))P(\overline{\mathcal{R}} \cap \mathcal{H}^{\mathcal{R}}(\mathbf{x}_\mathcal{A})) \\
&= P(\overline{\mathcal{R}} \cap \mathcal{H}^{\mathcal{R}}(\mathbf{x}_\mathcal{A}))(P(\mathcal{R} \cap \mathcal{H}^{\mathcal{R}}(\mathbf{x}_\mathcal{A})) + P(\overline{\mathcal{R}} \cap \mathcal{H}^{\mathcal{R}}(\mathbf{x}_\mathcal{A}))) \\
&= \left( 1 - \prod_{i \in (\mathcal{R} \cap \mathcal{A})} \mathbb{I}(X_i = 1) \prod_{j \in (\mathcal{R} \setminus \mathcal{A})} \theta_j \right) \left( \prod_{k \in \mathcal{R} \cap \mathcal{A}} \theta_k^{\mathbf{x}_\mathcal{A}(k)} (1 - \theta_k)^{1 - \mathbf{x}_\mathcal{A}(k)} \right)^2
\end{aligned}
\tag{8}
$$

Using (7) and (8) we can express the $f_{\text{EC}}(\mathbf{x}_\mathcal{A})$ as

$$
\begin{aligned}
f_{\text{EC}}(\mathbf{x}_\mathcal{A}) &= 1 - \frac{w_{\text{EC}}(\{\mathcal{S}_i\} \cap \mathcal{H}^{\mathcal{R}}(\mathbf{x}_\mathcal{A}))}{w_{\text{EC}}(\{\mathcal{S}_i\})} \\
&= 1 - \frac{\left( 1 - \prod_{i \in (\mathcal{R} \cap \mathcal{A})} \mathbb{I}(X_i = 1) \prod_{j \in (\mathcal{R} \setminus \mathcal{A})} \theta_j \right) \left( \prod_{k \in \mathcal{R} \cap \mathcal{A}} \theta_k^{\mathbf{x}_\mathcal{A}(k)} (1 - \theta_k)^{1 - \mathbf{x}_\mathcal{A}(k)} \right)^2}{1 - \prod_{i \in \mathcal{R}} \theta_i}
\end{aligned}
\tag{9}
$$

**Lemma 1.** *The expression $f_{\text{EC}}(\mathbf{x}_\mathcal{A})$ is strongly adaptive monotone and adaptive submodular.*

*Proof.* See Appendix A $\qquad\qquad\qquad\qquad\qquad\qquad\qquad\qquad\qquad\qquad\qquad\qquad\square$

### 4.3 Improvement in runtime from exponential to linear

For non-uniform priors, computing (5) is difficult. The naive approach is to compute all hypothesis and assign them to correct subregions and then compute the weights. This has a runtime of a runtime of $\mathcal{O}\left(2^{\mathcal{T}}\right)$.

However, our expression (9) can be computed in $\mathcal{O}\left(\mathcal{T}\right)$. This is because of the simplifications induced by the independent bernoulli assumption.

Since we have to repeat this computation every iteration of the algorithm, we can reduce this to $\mathcal{O}\left(1\right)$ through memoization. If we memoize $\left( 1 - \prod_{i \in (\mathcal{R} \cap \mathcal{A})} \mathbb{I}(X_i = 1) \prod_{j \in (\mathcal{R} \setminus \mathcal{A})} \theta_j \right)$, we can incrementally update it every time a test $t$ is evaluated. We also need to memoize $\left( \prod_{k \in \mathcal{R} \cap \mathcal{A}} \theta_k^{\mathbf{x}_\mathcal{A}(k)} (1 - \theta_k)^{1 - \mathbf{x}_\mathcal{A}(k)} \right)^2$ and update it incrementally.

### 4.4 Solving the original DRD problem using BISECT

We now return to the Bern-DRD (2) where we have multiple regions $\{\mathcal{R}_1, \ldots, \mathcal{R}_m\}$ that can overlap and the goal is to push the probability into one such region. Similar to DIRECT (Chen et al. [5]), we apply BISECT to solve the problem.

#### 4.4.1 The Noisy-OR Construction

The general strategy is to reduce the DRD problem with $m$ regions to $O(m)$ instances of the ECD problem such that *solving any one of them* is sufficient for solving the DRD problem as shown in Fig. 4.

ECD problem $r$ creates a 'one region versus all' problem using $\mathcal{R}_r$. The EC$^2$ objective corresponding to this problem is $f_{\text{EC}}^r(\mathbf{x}_\mathcal{A})$. Note that $f_{\text{EC}}^r(\emptyset) = 0$ which corresponds to nothing. On the other hand $f_{\text{EC}}^r(\mathbf{x}_\mathcal{T}) = 1$ which implies all edges are cut. The DIRECT algorithm then combines them in a *Noisy-OR* formulation by defining the following combined objective

$$
f_{\text{DRD}}(\mathbf{x}_\mathcal{A}) = 1 - \prod_{r=1}^m (1 - f_{\text{EC}}^r(\mathbf{x}_\mathcal{A}))
\tag{10}
$$

Figure 4: The DRD problem split into 'one region versus all' ECD problems by the DIRECT algorithm

Note that $f_{\text{DRD}}(\mathbf{x}_{\mathcal{A}}) = 1$ iff $f_{\text{EC}}^r(\mathbf{x}_{\mathcal{A}}) = 1$ for at least one $r$. Thus the original DRD problem (2) is equivalent to solving

$$\pi^* \in \underbrace{\arg\min_{\pi} c(\pi)}_{\text{find policy}} \text{ s.t } \underbrace{\forall \mathbf{x}_{\mathcal{T}}}_{\text{groundtruth}} : \underbrace{f_{\text{DRD}}(\mathbf{x}_{\mathcal{A}}(\pi, \mathbf{x}_{\mathcal{T}})) \geq 1}_{\text{drive the objective to 1}} \tag{11}$$

DIRECT greedily selects a test $t^* \in \arg\max_t \frac{\Delta_{f_{\text{DRD}}}(t \mid \mathbf{x}_{\mathcal{A}})}{c(t)}$.

### 4.4.2 The BISECT algorithm

We can now evaluate the DRD objective in (10) using (9)

$$f_{\text{DRD}}(\mathbf{x}_{\mathcal{A}})$$

$$= 1 - \prod_{r=1}^{m} (1 - f_{\text{EC}}^r(\mathbf{x}_{\mathcal{A}}))$$

$$= 1 - \prod_{r=1}^{m} \left( 1 - 1 + \frac{\left(1 - \prod_{i \in (\mathcal{R}_r \cap \mathcal{A})} \mathbb{I}(X_i = 1) \prod_{j \in (\mathcal{R}_r \setminus \mathcal{A})} \theta_j \right) \left( \prod_{k \in \mathcal{R}_r \cap \mathcal{A}} \theta_k^{\mathbf{x}_{\mathcal{A}}(k)} (1 - \theta_k)^{1 - \mathbf{x}_{\mathcal{A}}(k)} \right)^2}{1 - \prod_{i \in \mathcal{R}_r} \theta_i} \right)$$

$$= 1 - \prod_{r=1}^{m} \left( \frac{\left(1 - \prod_{i \in (\mathcal{R}_r \cap \mathcal{A})} \mathbb{I}(X_i = 1) \prod_{j \in (\mathcal{R}_r \setminus \mathcal{A})} \theta_j \right) \left( \prod_{k \in \mathcal{R}_r \cap \mathcal{A}} \theta_k^{\mathbf{x}_{\mathcal{A}}(k)} (1 - \theta_k)^{1 - \mathbf{x}_{\mathcal{A}}(k)} \right)^2}{1 - \prod_{i \in \mathcal{R}_r} \theta_i} \right)$$

$$\tag{12}$$

**Lemma 2.** *The expression $f_{\text{DRD}}(\mathbf{x}_{\mathcal{A}})$ is strongly adaptive monotone and adaptive submodular.*

*Proof.* See Appendix B ◻

**Theorem 1.** *Let $m$ be the number of regions, $p_{\min}^h$ the minimum prior probability of any hypothesis, $\pi_{DRD}$ be the greedy policy and $\pi^*$ with the optimal policy. Then $c(\pi_{DRD}) \leq c(\pi^*)(2m \log \frac{1}{p_{\min}^h} + 1)$.*

*Proof.* See Appendix C ◻

We now describe the algorithm BISECT. Algorithm 1 shows the framework for a general decision region determination algorithm. In order to specify BISECT, we need to define two options - a candidate test set selection function `SelectCandTestSet`$(\mathbf{x}_{\mathcal{A}})$ and a test selection function `SelectTest`$(\mathcal{T}_{\text{cand}}, \boldsymbol{\theta}, \mathbf{x}_{\mathcal{A}})$.

The vanilla version of BISECT implements `SelectCandTestSet`$(\mathbf{x}_{\mathcal{A}})$ to return the set of all candidate tests $\mathcal{T}_{\text{cand}}$ that contains only tests belonging to active regions that have not already been evaluated

**Algorithm 1:** Decision Region Determination with Independent Bernoulli Test($\{\mathcal{R}_i\}_{i=1}^m, \boldsymbol{\theta}, \mathbf{x}_\mathcal{T}$)

1  $\mathcal{A} \leftarrow \emptyset$ ;
2  **while** ($\nexists \mathcal{R}_i, P(\mathcal{R}_i|\mathbf{x}_\mathcal{A}) = 1$) **and** ($\exists \mathcal{R}_i, P(\mathcal{R}_i|\mathbf{x}_\mathcal{A}) > 0$) **do**
3     $\mathcal{T}_{\text{cand}} \leftarrow \texttt{SelectCandTestSet}(\mathbf{x}_\mathcal{A})$ ;              ▷ Using either (13) or (15)
4     $t^* \leftarrow \texttt{SelectTest}(\mathcal{T}_{\text{cand}}, \boldsymbol{\theta}, \mathbf{x}_\mathcal{A})$ ;       ▷ Using either (14),(16),(17),(18) or (19)
5     $\mathcal{A} \leftarrow \mathcal{A} \cup t^*$;
6     $x_{t^*} \leftarrow \mathbf{x}_\mathcal{T}(t^*)$ ;                    ▷ Observe outcome for selected test

| Test | Bias | Gain |
|------|------|------|
| $[t]$ | $[\theta_t]$ | $[\Delta_{f_{\text{DRD}}}(t)]$ |
| 1 | 0.8 | 0.075 |
| 2 | 0.7 | 0.093 |
| 3 | 0.8 | 0.070 |
| 4 | 0.8 | 0.070 |
| 5 | 0.6 | 0.113 |

| Test | Bias | Gain |
|------|------|------|
| $[t]$ | $[\theta_t]$ | $[\Delta_{f_{\text{DRD}}}(t)]$ |
| 1 | 0.8 | 0.161 |
| 2 | 0.7 | 0.199 |
| 3 | 0.8 | 0.132 |
| 4 | 0.8 | 0.132 |
| 5 | 0.3 | 0.180 |

(a)                                                   (b)

Figure 5: Canonical example illustrating BISECT. In both scenarios, the paths remain the same but the bias vector $\boldsymbol{\theta}$ varies. (a) Test 5, which is common to 2 paths. $\theta_5 = 0.6$ implies that 5 is an informative test as its outcome not only affects the probability of a lot of paths, but it also has a slight likelihood of being collision free. Hence its gain is 0.113. (b) Setting $\theta_5 = 0.3$ reduces the likelihood of the test being true. Hence its no longer informative and instead test 2 with gain 0.199 is chosen.

$$\mathcal{T}_{\text{cand}} = \left\{ \bigcup_{i=1}^m \{\mathcal{R}_i \mid P(\mathcal{R}_i|\mathbf{x}_\mathcal{A}) > 0\} \right\} \setminus \mathcal{A} \tag{13}$$

We now examine the BISECT test selection rule $\texttt{SelectTest}(\mathcal{T}_{\text{cand}}, \boldsymbol{\theta}, \mathbf{x}_\mathcal{A})$ which can be simplified as

$$\begin{aligned}
t^* &\in \underset{t \in \mathcal{T}_{\text{cand}}}{\arg\max} \frac{\Delta_{f_{\text{DRD}}}(t \mid \mathbf{x}_\mathcal{A})}{c(t)} \\
&\in \underset{t \in \mathcal{T}_{\text{cand}}}{\arg\max} \frac{\mathbb{E}_{x_t}\left[f_{\text{DRD}}(\mathbf{x}_{\mathcal{A}\cup\{t\}}) - f_{\text{DRD}}(\mathbf{x}_\mathcal{A}) \mid \mathbf{x}_\mathcal{A}\right]}{c(t)} \\
&\in \underset{t \in \mathcal{T}_{\text{cand}}}{\arg\max} \frac{1}{c(t)} \mathbb{E}_{x_t} \Bigg[ \prod_{r=1}^m \left( 1 - \prod_{i\in(\mathcal{R}_r\cap\mathcal{A})} \mathbb{I}(X_i = 1) \prod_{j\in(\mathcal{R}_r\setminus\mathcal{A})} \theta_j \right)
\end{aligned} \tag{14}$$
$$- \left( \prod_{r=1}^m \left( 1 - \prod_{i\in(\mathcal{R}_r\cap\mathcal{A}\cup t)} \mathbb{I}(X_i = 1) \prod_{j\in(\mathcal{R}_r\setminus\mathcal{A}\cup t)} \theta_j \right) \right) \left(\theta_t^{x_t}(1-\theta_t)^{1-x_t}\right)^{2\sum_{k=1}^m \mathbb{I}(t\in\mathcal{R}_k)} \Bigg]$$

Fig. 5 illustrates how BISECT chooses different tests dependent on the bias vector $\boldsymbol{\theta}$.

We now discuss the complexity of computing the marginal gain at each iteration. We have to cycle through $n$ tests. For each tests, we only have to cycle through regions which it impacts. Let $\eta$ be the maximum number of regions that any test belongs to. For every region, we need to do an $O(1)$ operation of calculating the change in probability. Hence the complexity is $O(n\eta)$. Note that this can be faster in practice by leveraging lazy methods in adaptive submodular problems (Golovin and Krause [16]).

### 4.5   Adaptively constraining test selection to most likely region

We observe in our experiments that the surrogate (12) suffers from a slow convergence problem - $f_{\text{DRD}}(\mathbf{x}_\mathcal{A})$ takes a long time to converge to 1 when greedily optimized. This can be attributed to the curvature of the function. To alleviate the convergence problem, we introduce an alternate candidate selection function $\texttt{SelectCandTestSet}(\mathbf{x}_\mathcal{A})$ that assigns to $\mathcal{T}_{\text{cand}}$ the set of all tests that belong to the most likely region $\mathcal{T}_{\text{maxP}}$. We hence forth denote the constraint as MAXPROBREG. It is evaluated

as follows

$$\mathcal{T}_{\mathrm{maxP}} = \left\{ \underset{\mathcal{R}_i=(\mathcal{R}_1,\mathcal{R}_2,...,\mathcal{R}_m)}{\arg\max} P(\mathcal{R}_i|\mathbf{x}_\mathcal{A}) \right\} \setminus \mathcal{A} \tag{15}$$

Applying the constraint in (15) leads to a dramatic improvement for any test selection policy as we will show in Sec. 6.7. The following theorem offers a partial explanation

**Theorem 2.** *A policy that greedily latches to a region according the the posterior conditioned on the region outcomes has a near-optimality guarantee of 4 w.r.t the optimal region evaluation sequence.*

*Proof.* See Appendix D □

Applying the constraint in (15) implies we are no longer greedily optimizing $f_{\mathrm{DRD}}(\mathbf{x}_\mathcal{A})$. However, the following theorem bounds the sub-optimality of this policy.

**Theorem 3.** *Let $p_{\min} = \min_i P(\mathcal{R}_i)$, $p_{\min}^h = \min_{h\in\mathcal{H}} P(h)$ and $l = \max_i |\mathcal{R}_i|$. The policy using (15) has a suboptimality of $\alpha\left(2m\log\left(\frac{1}{p_{\min}^h}\right) + 1\right)$ where $\alpha \le \left(1 - \max\left((1-p_{\min})^2, p_{\min}^{\frac{2}{l}}\right)\right)^{-1}$.*

*Proof.* See Appendix E □

The complexity of BISECT with MAXPROBREG reduces since we only have to visit states belonging to the most probable path. Finding the most probable path is an $O(m)$ operation. Let $l$ be the maximum number of tests in a region. Hence the complexity of gain calculation is $O(l\eta)$. The total complexity is $O(l\eta + m)$.

## 5 Heuristic approaches to solving Bernoulli DRD problem

We propose a collection of competitive heuristics that can also be used to solve the Bern-DRD problem. These heuristics are various $\texttt{SelectTest}(\mathcal{T}_{\mathrm{cand}}, \boldsymbol{\theta}, \mathbf{x}_\mathcal{A})$ policies in the framework of Alg. 1. To simplify the setting, we assume unit cost $c(t) = 1$ although it would be possible to extend these to nonuniform setting. We also state the complexity for each algorithm and summarize them in Table 1.

### 5.1 RANDOM

The first heuristic RANDOM selects a test by sampling uniform randomly

$$t^* \in \mathcal{T}_{\mathrm{cand}} \tag{16}$$

The complexity is $O(1)$.

### 5.2 MAXTALLY

We adopt our next heuristic MAXTALLY from Dellin and Srinivasa [11] by where the test belonging to most regions is selected. This criteria exhibits a 'fail-fast' characteristic where the algorithm is incentivized to eliminate options quickly. This policy is likely to do well where regions have large amounts of overlap on tests that are likely to be in collision.

$$t^* \in \underset{t\in\mathcal{T}_{\mathrm{cand}}}{\arg\max} \sum_{i=1}^{m} \mathbb{I}\left(t \in \mathcal{R}_i, P(\mathcal{R}_i|\mathbf{x}_\mathcal{A}) > 0\right) \tag{17}$$

To evaluate the complexity, we first describe how to efficiently implement this algorithm. Note that we can pre-process regions and tests to create a tally count of tests belonging to regions and a reverse lookup from tests to regions. Hence selecting a tests is simply finding the test with the max tally which is $O(n)$. If the test is in collision, the tally count is updated by looking at all regions the test affects, and visiting tests contained by those regions to reduce their tally count. Let $\eta$ be the maximum regions to which a test belongs, and $l$ be the maximum number of tests contained by a region. Hence the complexity is $O(n + \eta l)$. In the MAXPROBREG setting, the complexity reduces to $O(l + \eta l) = O((1+\eta)l)$.

Table 1: Complexity of different algorithms (number of tests $n$, number of regions $m$, maximum tests in a region $l$ and maximum regions belonging to a test $\eta$ )

|  | MVoI | Random | MaxTally | SetCover | BiSect |
|---|---|---|---|---|---|
| Unconstrained |  | $O(1)$ | $O(n + \eta l)$ | $O(n^2 m)$ | $O(n\eta)$ |
| MaxProbReg | $O(\eta l)$ | $O(1)$ | $O((1 + \eta)l)$ | $O(lmn)$ | $O(\eta l + m)$ |

## 5.3 SetCover

The next policy SetCover selects tests that maximize the expected number of 'covered' tests, i.e. if a test is in collision, how many more tests are eliminated.

$$t^* \in \argmax_{t \in \mathcal{T}_{\text{cand}}} (1-\theta_t) \left| \left\{ \bigcup_{i=1}^{m} \{\mathcal{R}_i \mid P(\mathcal{R}_i|\mathbf{x}_\mathcal{A}) > 0\} - \bigcup_{j=1}^{m} \{\mathcal{R}_j \mid P(\mathcal{R}_j|, {}_{X_t=0}^{\mathbf{x}_\mathcal{A},}) > 0\} \right\} \setminus \{\mathcal{A} \cup \{t\}\} \right| \tag{18}$$

The motivation for this policy has its roots in the question - what is the optimal policy for checking *all* paths? While Bern-DRD requires identifying one feasible region, it might still benefit from such a policy in situations where only one region is feasible. The following theorem states that greedily selecting tests according to the criteria above has strong guarantees.

**Theorem 4.** SetCover *is a near-optimal policy for the problem of optimally checking all regions.*

*Proof.* See Appendix F ☐

We now analyze the complexity. We have to visit every test. Given a test is in collision, we have to compute the number of tests in the remaining regions which are not invalid. This would require visiting every test in every region. Hence the complexity is $O(n^2 m)$. In the MaxProbReg setting, the complexity reduces to $O(lmn)$, where $l$ is the maximum number of tests contained by a region.

## 5.4 MVoI

The last baseline is a classic heuristic from decision theory: myopic value of information Howard [19]. We define a utility function $U(h, \mathcal{R}^\mathcal{H})$ which is 1 if $h \in \mathcal{R}^\mathcal{H}$ and 0 otherwise. The utility of $\mathcal{H}^\mathcal{R}(\mathbf{x}_\mathcal{A})$ corresponds to the maximum expected utility of any decision region, i.e., the expected utility if we made a decision now. MVoI greedily chooses the test that maximizes (in expectation over observations) the utility as shown.

$$t^* \in \argmax_{t \in \mathcal{T}_{\text{maxP}}} (1 - \theta_t) \max_{i=1,\ldots,m} P(\mathcal{R}_i \mid \mathbf{x}_\mathcal{A}, X_t = 0) \tag{19}$$

Note that this test selection works only in the MaxProbReg setting. For every test in the most probable region, we eliminate regions that would invalid if the test is invalid. Let $l$ be the maximum number of tests contained by a region. Let $\eta$ be the maximum number of regions contained by a test. Then the complexity is $O(\eta l)$.

# 6 Experiments

We evaluate all algorithms on a spectrum of synthetic problems, motion planning problems and experimental data from an autonomous helicopter. We present details on each dataset - motivation, construction of regions and tests and analysis of results. Table 2 presents the performance of all algorithms on all datasets. It shows the normalized cost with respect to algorithm BiSect $\pi_{f_{\text{DRD}}}$, i.e. $\frac{c(\pi)-c(\pi_{f_{\text{DRD}}})}{c(\pi_{f_{\text{DRD}}})}$. The 95% confidence interval value is shown (as a large number of samples are required to drive down the variance). Finally, in Section 6.7, we present a set of overall hypothesis and discuss their validity.

Figure 6: Construction of candidate path library $\Xi$ for synthetic GBG experiments. The paths are embedded in an underlying RGG. (a) 100 paths (b) 500 paths (c) 1000 paths

## 6.1 Dataset 1: Synthetic Bernoulli Test

### 6.1.1 Motivation

These datasets are designed to check the general applicability of our algorithms on problems which do not arise from graphs. Hence regions and tests are randomly created with minimal constraints that ensure the problems are non-trivial.

### 6.1.2 Construction

First, a boolean region to test allocation matrix $\mathbf{A} \in \{0,1\}^{m \times n}$ is created where $\mathbf{A}(i,j) = 1$ implies whether test $j$ belongs to region $\mathcal{R}_i$. $\mathbf{A}$ is randomly allocated by ensuring that each region $\mathcal{R}_i$ contains a random subset of tests. The number of such tests $l_i$ varies with region and is randomly sampled uniformly from $[0.05n, 0.10n]$. The bias vector $\boldsymbol{\theta} \in \mathbb{R}^{1 \times n}$ is sampled uniformly randomly from $[0.1, 0.9]$. A set of $N_{\text{test}}$ problems are created by sampling a ground truth $\mathbf{x}_{\mathcal{T}}$ from $\boldsymbol{\theta}$, and ensuring that at least one region is valid in each problem.

We set $n = 100$ and $N_{\text{test}} = 100$. We create 3 datasets by varying the number of regions $m = \{100, 500, 1000\}$. This is to investigate the performance of algorithms as the overlap among regions increase.

### 6.1.3 Analysis

Table 2 shows the results as regions are varied. Among the unconstrained algorithms, BISECT outperforms all other algorithms substantially with the gap narrowing on the Large dataset. For the MAXPROBREG versions, BISECT remains competitive across all datasets. MVOI matches its performance, doing better on dataset Large ($m = 1000$). From these results, we conclude that the datasets favour myopic behaviour. The performance of MVOI increases monotonically with $m$. This can be attributed to the fact that as the number of probable regions increase, myopic policies tend to perform better.

## 6.2 Dataset 2: Synthetic Generalized Binomial Graph

### 6.2.1 Motivation

These datasets are designed to test algorithms on GBG which do not necessarily arise out of motion planning problems. For these datasets, edge independence is directly enforced. Difference between results on these datasets and those from motion planning can be attributed to spatial distribution of obstacles and overlap among regions.

### 6.2.2 Construction

A randomg geometric graph (RGG) [28] $G = (V, E)$ with 100 vertices is sampled in a unit box $[0, 1] \times [0, 1]$. We create a set of paths $\Xi$ from this graph by solving a set of shortest path problems (SPP). In each iteration of this algorithm, edges from $G$ are randomly removed with probability 0.5 and the SPP is solved to produce $\xi$. This path is then appended to $\Xi$ (if already not in the set) until $|\Xi| = m$. A bias vector $\boldsymbol{\theta} \in \mathbb{R}^{1 \times |E|}$ is sampled uniformly randomly from $[0.1, 0.9]$.

Figure 7: Example of BISECT as applied to a synthetic GBG problem. The GBG is shown with edges colored from magenta (thin) to cyan (thick) according to the prior likelihoods. (a) Initial state of the problem (b) BISECT selects the most probable path and checks all its edges till it encounters and edge in collision in the middle of the path (c) It then looks at alternates till it discovers a valid short cut to connect the first and second half of the path

We create 3 datasets by varying the number of paths $m = \{100, 500, 1000\}$. For each dataset, we create $N_{\text{test}} = 100$ problems. Fig. 6 shows the paths for these datasets.

### 6.2.3 Analysis

Table 2 shows the results as the number of paths is increased. Among the unconstrained algorithms, MAXTALLY does better than BISECT when $m$ is small. As $m$ increases, BISECT outperforms all others and even matches up to its MAXPROBREG version. This can be attributed to the fact that when $m$ is small, most of the paths pass through 'bottleneck edges'. MAXTALLY inspects these edges first and if they are in collision, eliminates options quickly. As $m$ increases, the fraction of overlap decreases and problems become harder. For these problems, simply checking the most common edge does not suffice.

For the MAXPROBREG version, we see that MAXTALLY has better overall performance. Thus we conclude that the combination of checking the most common edge and constraining to the most probable path works well. The difference between these datasets and Section 6.1 is that the region test allocation appears naturally from the graph structure. This leads to problems where 'bottleneck edges' exist and MAXTALLY is able to identify them. Its interesting to note that MVOI performs worse as $m$ increases. This is because of the optimistic nature of MVOI- its less likely to select an edge that eliminates a lot of high probability regions (contrary to MAXTALLY). Hence the contrast between the two algorithms is displayed here. Fig. 7 shows an illustration of BISECT selecting edges to solve a problem for $m = 50$.

## 6.3 Dataset 3: 2D Geometric Planning

### 6.3.1 Motivation

The main motivation for our work is robotic motion planning. The simplest instantiation is 2D geometric planning. The objective is to plan on a purely geometric graph where edges are invalidated by obstacles in the environment. Hence the probability of collision appears from the chosen distribution of obstacles. While the independent Bernoulli assumption is not valid, we will see that the algorithms still leverage such a prior to make effective decisions.

### 6.3.2 Construction

A random geometric graph (RGG) [28] $G = (V, E)$ with $|V| = 200$ is sampled in a unit box $[0, 1] \times [0, 1]$. We define a world map $\mathcal{M}$ as a binary map of occupied and unoccupied cells. Given $G$ and a $\mathcal{M}$, and edge $e \in E$ is said to be in collision if it passes through an unoccupied cell. Fig. 8(a) shows an example of a collision checked RGG. A parametric distribution can be used to create a distribution over world maps $P(\mathcal{M})$ which defines different environments. $P(\mathcal{M})$ can be used to measure the probability of individual edges being in collision.

We create 3 datasets corresponding to different environments as shown in Fig. 9 - Forest, OneWall, TwoWall. These datasets are created by defining parametric distributions that distribute rectangular blocks. Forest corresponds to a non uniform stationary distribution of squares to mimic a forest like environment where trees are clustered together with spatial correlations. OneWall is created by constructing a wall with random gaps in conjunction with a uniform random distribution of squares.

Figure 8: Different explicit graphs for different problem settings (a) A RGG for 2D geometric planning (b) A state lattice for non-holonomic planning.

Figure 9: Different datasets of environments (a) OneWall (b) TwoWall (c) Forest

TwoWall contains two such walls. Hence these datasets create a spectrum of difficulty to test our algorithms.

We now describe the method for constructing the set of paths $\Xi$. We would like a set of good candidate paths on the distribution $P(\mathcal{M})$. We define a goodness function as the probability of atleast one path in the set to be valid on the dataset. Following the methodology in Tallavajhula et al. [30], we use a greedy method. We sample a training dataset consisting of $N_{\text{train}} = 1000$ problems. On every problem in this dataset, we solve the shortest path problem to get a path $\xi$. We then greedily construct $\Xi$ by selecting the path that is most valid till our budget $m$ is filled. We set $m = 500$ for all datasets.

### 6.3.3 Analysis

Table 2 shows the results on all 3 datasets. In the unconstrained case, BISECT outperforms all other algorithms by a significant margin. For the MAXPROBREG version, BISECT remains competitive. The closest competitor to it is SETCOVER- matching performance in the TwoWall dataset. Further analysis of this dataset revealed that the dataset has problems that are difficult - where only one of the paths in the set are feasible. This often requires eliminating all other paths. SETCOVER performs well under such situations due to guarantees described in Theorem 4.

These results vary from the patterns in Section 6.2. This is to do with the relationship with overlap of regions and priors on tests. Since the regions are created in a way cognizant of the prior, regions often overlap on tests that are likely to be free with high probability. MAXTALLY ignores this bias term and hence prioritizes checking such edges first even if they offer no information.

Table 2 also shows results on varying the number of regions. BISECT is robust to this change. SETCOVER performs better with less number of paths. This can be attributed to the path that the number of feasible path decreases, thus becoming advantageous to check all paths.

Fig. 10 shows a comparison of all algorithms on a problem from OneWall dataset. It illustrates the contrasting behaviours of all algorithms. MAXTALLY selects edges belonging to many paths which happens to be near the start / goal. These are less likely to be discriminatory. SETCOVER takes time to converge as it attempts to cover all edges. MVOI focuses on edges likely to invalidate the current most probable path which eliminates paths myopically but takes time to converge. BISECT enjoys the best of all worlds.

Figure 10: Performance (number of evaluated edges) of all algorithms on 2D geometric planning. Snapshots are at interim and final stages respectively show evaluated valid edges (green), invalid (red) edges and final path (magenta). The marginal gain of candidate edges goes from black (low) to cream (high).

## 6.4 Dataset 4: Non-holonomic Path Planning

### 6.4.1 Motivation

While 2D geometric planning examined the influence of various spatial distribution of obstacles on random graphs, it does not impose a constraint on the class of graphs. Hence we look at the more practical case of mobile robots with constrained dynamics. This robots plan on a state-lattice (Pivtoraiko et al. [29]) - a graph where edges are dynamically feasible maneuvers. As motivated in Section 1, these problems are of great importance as a robot has to react fast to safely avoid obstacles. The presence of differential constraint reduces the set of feasible paths, hence requiring checks at a greater resolution.

### 6.4.2 Construction

The vehicle being considered is a planar curvature constrained system. Hence the search space is 3D - x, y and yaw. A state lattice of dynamically feasible maneuvers is created as shown in Fig. 8(b). The environments are used from Section 6.3 - Forest and OneWall. The density of obstacle in these datasets are altered to allow constrained system to find solutions. The candidate set of paths are created in a similar fashion as in Section. 6.3. We set $m \approx 100$ for all datasets.

### 6.4.3 Analysis

Table 2 shows results across datasets. In the unconstrained setting, BISECT significantly outperforms other algorithms. In the MAXPROBREG setting, we see that SETCOVER is equally competitive. The analysis of the Forest dataset reveals that due to the difficulty of the dataset, problems are such that only one of the paths is free. As explained in Section 6.3, SETCOVER does well in such settings. On the OneWall dataset, we see several algorithms performing comparatively. This might indicate the relative easiness of the dataset.

Table 2 shows variation across degree of the lattice. We see that BISECT remains competitive across this variation.

Figure 11: 7D arm planning dataset (a) Snapshot of the manipulator for planning with a table (b) Snapshot of manipulator planning with an object (c) The explicit graph shown as straight line connections between end effector locations (also subsampled $50\%$). The start and goal end effector locations are also shown. Edges in collision are removed.

## 6.5 Dataset 5: 7D Arm Planning

### 6.5.1 Motivation

An important application for efficient edge evaluations is planning for a 7D arm. Edge evaluation is expensive geometric intersection operations are required to be performed to ascertain validity. A detailed motivation is provided in Dellin and Srinivasa [11]. Efficient collision checking would allow such systems to plan quickly while performing tasks such as picking up and placing objects from one tray to another. One can additionally assume an unknown agent present in the workspace. Such problems would benefit from reasoning using priors on edge validity.

### 6.5.2 Construction

A random geometric graph with 7052 vertices and 16643 edges is created (as described in Dellin and Srinivasa [11]). Edges in self-collision are prune apriori. We create 2 datasets to simulate pick and place tasks in a kitchen like environment. The start and goal from all problem is from one end-effector position to another. The first dataset - Table - comprises simply of a table at random offsets from the robot. The location of the table invalidates large number of edges. The second dataset - Clutter - comprises of an object and table at random offsets from the robot. In all datasets, a random subset corresponding to $0.3$ fraction of free edges are 'flipped', i.e. made to be in collision. This creates the effect of random disturbances in the environment. Paths are created in a similar way as Section 6.3. We set $m \approx 200$ for all datasets. Fig. 11 shows an illustration of the problems.

### 6.5.3 Analysis

Table 2 shows results across datasets. In both the unconstrained and MAXPROBREG setting, BISECT significantly outperforms other algorithms. MAXTALLY in the MAXPROBREG is the next best performing policy. This suggests that the dataset might lead to bottleneck edges - edges through which many paths pass through that can be in collision. Further analysis reveals, this artifact occurs due to the random disturbance. MAXTALLY is able to verify quickly if such bottleneck edges are in collision, and if so remove a lot of candidate paths from consideration.

## 6.6 Autonomous Helicopter Wire Avoidance

We now evaluate our algorithms on experimental data from a full scale helicopter. The helicopter is equipped with a laser scanner that scans the world to build a model of obstacles and free space. The system is required to plan around detected obstacles as it performs various missions.

A particularly difficult problem is dealing with wires as the system comes in to land. The system has limitations on how fast it can ascend / descend. Hence it has to not only react fast, but determine which direction to move so as to feasibly land. Fig. 12 shows the scenario. In this domain, edge evaluation is expensive because given an edge, it must be checked at a high resolution to ensure it is as sufficient distance from an obstacle.

Fig. 12 (b) shos how BISECT evaluates informative edges to identify a feasible path. This algorithm uses priors collected in simulation of wire like environments.

Table 2: Normalized cost of different algorithms on different datasets (95% C.I.)

| | **MVoI** | **RANDOM** Unconstrained MaxProbReg | **MAXTALLY** Unconstrained MaxProbReg | **SETCOVER** Unconstrained MaxProbReg | **BISECT** Unconstrained MaxProbReg |
|---|---|---|---|---|---|
| **Synthetic Bernoulli Test: Variation across region overlap** | | | | | |
| Small | | (4.18, 6.67) | (3.49, 5.23) | (1.77, 3.01) | (1.42, 2.36) |
| $m:100$ | (0.00, 0.08) | (0.12, 0.29) | (0.12, 0.25) | (0.18, 0.40) | (0.00, 0.00) |
| Medium | | (3.27, 4.40) | (3.04, 4.30) | (3.55, 4.67) | (1.77, 2.64) |
| $m:500$ | (0.00, 0.00) | (0.05, 0.25) | (0.14, 0.24) | (0.14, 0.33) | (0.00, 0.00) |
| Large | | (2.86, 4.26) | (2.62, 3.85) | (2.94, 3.71) | (1.33, 1.81) |
| $m:1000$ | (−0.11, 0.00) | (0.00, 0.28) | (0.06, 0.26) | (0.09, 0.22) | (0.00, 0.00) |
| **Synthetic Bernoulli Test: Variation across region overlap** | | | | | |
| Small | | (6.08, 7.25) | (0.68, 1.50) | (2.12, 2.50) | (1.27, 1.50) |
| $m:100$ | (0.00, 0.00) | (0.00, 0.00) | (−0.13, −0.11) | (0.13, 0.14) | (0.00, 0.00) |
| Medium | | (6.51, 8.53) | (0.12, 0.51) | (1.43, 1.75) | (0.15, 0.46) |
| $m:500$ | (0.00, 0.11) | (0.00, 0.11) | (0.00, 0.09) | (−0.04, 0.07) | (0.00, 0.00) |
| Large | | (9.65, 11.67) | (0.63, 1.18) | (2.24, 2.89) | (0.31, 0.63) |
| $m:1000$ | (0.13, 0.24) | (0.00, 0.11) | (−0.13, −0.07) | (0.11, 0.13) | (0.00, 0.00) |
| **2D Geometric Planning: Variation across environments** | | | | | |
| Forest | | (19.45, 27.66) | (4.68, 6.55) | (3.53, 5.07) | (1.90, 2.46) |
| | (0.03, 0.18) | (0.13, 0.30) | (0.09, 0.18) | (0.00, 0.09) | (0.00, 0.00) |
| OneWall | | (13.35, 17.79) | (4.12, 4.89) | (1.36, 2.11) | (0.76, 1.20) |
| | (0.045, 0.21) | (0.11, 0.42) | (0.00, 0.12) | (0.14, 0.29) | (0.00, 0.00) |
| TwoWall | | (13.76, 16.61) | (2.76, 3.93) | (2.07, 2.94) | (0.91, 1.44) |
| | (0.00, 0.09) | (0.33, 0.51) | (0.10, 0.20) | (0.00, 0.00) | (0.00, 0.00) |
| **2D Geometric Planning: Variation across region size** | | | | | |
| OneWall | | (12.06, 16.01) | (4.47, 5.13) | (2.00, 3.41) | (0.94, 1.42) |
| $m:300$ | (0.00, 0.17) | (0.12, 0.42) | (0.06, 0.24) | (0.00, 0.38) | (0.00, 0.00) |
| OneWall | | (13.26, 16.79) | (2.18, 3.77) | (1.04, 1.62) | (0.41, 0.91) |
| $m:858$ | (0.00, 0.14) | (0.09, 0.27) | (−0.04, 0.08) | (0.00, 0.14) | (0.00, 0.00) |
| **Non-holonomic Path Planning: Variation across environments** | | | | | |
| Forest | | (22.38, 29.67) | (9.79, 11.14) | (2.63, 5.28) | (1.54, 2.46) |
| | (0.09, 0.18) | (0.46, 0.79) | (0.25, 0.38) | (0.00, 0.00) | (0.00, 0.00) |
| OneWall | | (13.02, 15.75) | (8.40, 11.47) | (3.72, 4.54) | (3.28, 3.78) |
| | (−0.11, 0.11) | (0.00, 0.12) | (0.21, 0.28) | (−0.11, 0.11) | (0.00, 0.00) |
| **Non-holonomic Path Planning: Variation across lattice degree** | | | | | |
| OneWall | | (10.46, 11.57) | (3.95, 4.83) | (0.83, 1.18) | (0.24, 0.58) |
| $k:12$ | (0.04, 0.11) | (0.30, 0.56) | (0.11, 0.18) | (0.00, 0.06) | (0.00, 0.00) |
| OneWall | | (14.97, 17.90) | (9.19, 13.11) | (3.22, 5.07) | (2.16, 2.81) |
| $k:30$ | (0.05, 0.10) | (0.14, 0.40) | (0.20, 0.52) | (0.00, 0.03) | (0.00, 0.00) |
| **7D Arm Planning: Variation across environments** | | | | | |
| Table | | (15.12, 19.41) | (4.80, 6.98) | (1.36, 2.17) | (0.32, 0.67) |
| | (0.28, 0.54) | (0.13, 0.31) | (0.00, 0.04) | (0.00, 0.11) | (0.00, 0.00) |
| Clutter | | (7.92, 9.85) | (3.96, 6.44) | (1.42, 2.07) | (1.23, 1.75) |
| | (0.02, 0.20) | (0.14, 0.36) | (0.00, 0.00) | (0.00, 0.11) | (0.00, 0.00) |

Figure 12: Experimental data from a full scale helicopter that has to avoid wires as it comes into land. The helicopter detects wires fairly late which requires an instant avoidance maneuver. The helicopter uses a state lattice and has to quickly identify a feasible path on the lattice. Evaluating edges are expensive since the system has to ensure it avoids wires by a sufficient clearance. (a) A top down view of the state lattice. Maneuvers are lateral as well as vertical. (b) Performance of BISECT on the motion planning problem. The voxels in blue represent occupied locations in the world as detected by the helicopter. The wires (as seen in the camera) appear as a small set of voxels in the map. BISECT selectively evaluates certain edges of the state lattice (green shows edges evaluated to be valid, red shows edges evaluated to be invalid). It is quickly able to identify a feasible path.

Figure 13: (a) Illustration of convergence issues for $f_{\mathrm{DRD}}(\mathbf{x}_{\mathcal{A}})$ - the transformation $(1 - f_{\mathrm{DRD}}(\mathbf{x}_{\mathcal{A}}))^{\frac{1}{m}}$ shows that it flattens out thus allowing even a non-greedy algorithm to converge faster. (b) BISECT with MAXPROBREG shown in the space of posterior probabilities of region. First $\mathcal{R}_1$ is checked, then $\mathcal{R}_2$ and finally $\mathcal{R}_4$ is found to be valid.

## 6.7 Overall summary of results

Table 2 shows the evaluation cost of all algorithms on various datasets normalized w.r.t BISECT. The two numbers are lower and upper $95\%$ confidence intervals - hence it conveys how much fractionally poorer are algorithms w.r.t BISECT. The best performance on each dataset is highlighted. We present a set of observations to interpret these results.

**O 1.** BISECT *has a consistently competitive performance across all datasets.*

Table 2 shows on 13 datasets, BISECT is at par with the best - on 8 of those it is exclusively the best.

**O 2.** *The* MAXPROBREG *variant improves the performance of all algorithms on most datasets*

Table 2 shows that this is true on 12 datasets. The impact is greatest on RANDOM where improvement is upto a factor of 20. For the case of BISECT, Fig. 13(a) illustrates the problem by examining the shape of $(1 - f_{\mathrm{DRD}}(\mathbf{x}_{\mathcal{A}}))^{\frac{1}{m}}$. Even though $f_{\mathrm{DRD}}(\mathbf{x}_{\mathcal{A}})$ is submodular, it flattens drastically allowing a non-greedy policy to converge faster. Fig. 13(b) shows how the probability of region evolves as tests are checked in the MAXPROBREG setting. We see this 'latching' characteristic - where test selection drives a region probability to 1 instead of exploring other tests.

However, this is not true in general. See Appendix G for results on datasets with large disparity in region sizes.

**O 3.** *On planning problems,* BISECT *strikes a trade-off between the complimentary natures of* MAXTALLY *and* MVOI.

We examine this in the context of 2D planning as shown in Fig. 10. MAXTALLY selects edges belonging to many paths which is useful for path elimination but does not reason about the event when the edge is not in collision. MVOI selects edges to eliminate the most probable path but does not reason about how many paths a single edge can eliminate. BISECT switches between these behaviors thus achieving greater efficiency than both heuristics.

**O 4.** BISECT *checks informative edges in collision avoidance problems encountered a helicopter*

Fig. 12(b) shows the efficacy of BISECT on experimental flight data from a helicopter avoiding wire.

## 7    Conclusion

In this paper, we addressed the problem of identification of a feasible path from a library while minimizing the expected cost of edge evaluation given priors on the likelihood of edge validity. We showed that this problem is equivalent to a decision region determination problem where the goal is to select tests (edges) that drive uncertainty into a single decision region (a valid path). We proposed BISECT, and efficient and near-optimal algorithm that solves this problem by greedily optimizing a surrogate objective.We validated BISECT on a spectrum of problems against state of the art heuristics and showed that it has a consistent performance across datasets. This works serves as a first step towards importing Bayesian active learning approaches into the domain of motion planning.

## Footnotes

[1]Generally, edges in this graph are correlated, as edges in collision are likely to have neighbours in collision. Unfortunately, even measuring this correlation is challenging, especially in the high-dimensional non-linear configuration space of robot arms. Assuming independent edges is a common simplification [24, 26, 7, 2, 11]

[2]It is assumed that $c(e)$ is modular and non-zero. It can scale with edge length.

[3]Refer to supplementary on various methods to construct a library of good candidate paths

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

# Appendices

## Appendix A    Proof of Lemma 1

**Lemma.** *The expression $f_{\mathrm{EC}}(\mathbf{x}_{\mathcal{A}})$ is strongly adaptive monotone and adaptive submodular.*

*Proof.* The proof for $f_{\mathrm{EC}}(\mathbf{x}_{\mathcal{A}})$ is a straight forward application of Lemma 5 from Golovin et al. [17]. We now adapt the proof of adaptive submodularity from Lemma 6 in Golovin et al. [17]

We first prove the result for uniform prior. To prove adaptive submodularity, we must show that for all $\mathbf{x}_{\mathcal{A}} < \mathbf{x}_{\mathcal{B}}$ and $t \in \mathcal{T}$ , we have $\Delta_{\mathrm{EC}}\left(t \mid \mathbf{x}_{\mathcal{A}}\right) \geq \Delta_{\mathrm{EC}}\left(t \mid \mathbf{x}_{\mathcal{B}}\right)$. Fix $t$ and $\mathbf{x}_{\mathcal{A}}$, and let $\mathcal{V}(\mathbf{x}_{\mathcal{A}}) = \{h \mid P(h|\mathbf{x}_{\mathcal{A}}) > 0\}$ denote the version space, if $\mathbf{x}_{\mathcal{A}}$ encodes the observed outcomes. Let $n_{\mathcal{V}} = |\mathcal{V}(\mathbf{x}_{\mathcal{A}})|$ be the number of hypotheses in the version space. Likewise, let $n_{i,a}(\mathbf{x}_{\mathcal{A}}) = \left|\left\{h : h \in \mathcal{V}(\mathbf{x}_{\mathcal{A}}, X_t = a) \cap \mathcal{H}_i^{\mathcal{R}}\right\}\right|$, and let $n_a(\mathbf{x}_{\mathcal{A}}) = \sum\limits_{i=1}^{l} n_{i,a}(\mathbf{x}_{\mathcal{A}})$. We define a function $\phi$ of the quantities $n_{i,a} : 1 \leq i \leq l, a \in \{0, 1\}$ such that $\Delta_{\mathrm{EC}}\left(t \mid \mathbf{x}_{\mathcal{A}}\right) = \phi(n(\mathbf{x}_{\mathcal{A}}))$, where $n(\mathbf{x}_{\mathcal{A}})$ is the vector consisting of $n_{i,a}(\mathbf{x}_{\mathcal{A}})$ for all $i$ and $a$. For brevity, we suppress the dependence of $\mathbf{x}_{\mathcal{A}}$ where it is unambiguous.

It will be convenient to define $e_a$ to be the number of edges cut by $t$ such that at $t$ both hypotheses agree with each other but disagree with the realized hypothesis $h*$, conditioning on $X_t = a$. Written as a function of $n$, we have $e_a = \sum\limits_{i<j} \sum\limits_{b\neq a} n_{i,b}n_{j,b}$.

We also define $\gamma_a$ to be the number of edges cut by $t$ corresponding to self-edges belonging to hypotheses that disagree with the realized hypothesis $h^*$, conditioning on $X_t = a$. Written as a function of $n$, we have $\gamma_a = \sum\limits_{i} \sum\limits_{b\neq a} n_{i,b}^2$.

$$\phi(\mathbf{n}) = \sum_{i<j}\sum_{a\neq b} n_{i,a}n_{j,b} + \sum_a e_a\left(\frac{n_a}{n_{\mathcal{V}}}\right) + \sum_a \gamma_a\left(\frac{n_a}{n_{\mathcal{V}}}\right) \tag{20}$$

where $e_a = \sum\limits_{i<j}\sum\limits_{b\neq a} n_{i,b}n_{j,b}$ and $\gamma_a = \sum\limits_{i}\sum\limits_{b\neq a} n_{i,b}^2$. Here, $i$ and $j$ range over all class indices, and $a$ and $b$ range over all possible outcomes of test $t$. The first term on the right-hand side counts the number of edges that will be cut by selecting test $t$ no matter what the outcome of $t$ is. Such edges consist of hypotheses that disagree with each other at $t$ and, as with all edges, lie in different classes. The second term counts the expected number of edges cut by $t$ consisting of hypotheses that agree with each other at $t$. Such edges will be cut by $t$ iff they disagree with $h^*$ at $t$. The third term counts the expected number of edges cut by $t$ consisting of hypothesis with self-edges that disagree with $h^*$ at $t$.

We need to show $\frac{\partial\phi}{\partial n_{k,c}} \geq 0$ according to proof of Lemma 6 in Golovin et al. [17].

$$\frac{\partial\phi}{\partial n_{k,c}} = \frac{\partial}{\partial n_{k,c}}\left(\sum_{i<j}\sum_{a\neq b} n_{i,a}n_{j,b}\right) + \frac{\partial}{\partial n_{k,c}}\left(\sum_a e_a\left(\frac{n_a}{n_{\mathcal{V}}}\right)\right) + \frac{\partial}{\partial n_{k,c}}\left(\sum_a \gamma_a\left(\frac{n_a}{n_{\mathcal{V}}}\right)\right) \tag{21}$$

Expanding the first term in (21)

$$\frac{\partial}{\partial n_{k,c}}\left(\sum_{i<j}\sum_{a\neq b} n_{i,a}n_{j,b}\right) = \sum_{i\neq k, a\neq c} n_{i,a} \tag{22}$$

Expanding the second term in (21)

$$\frac{\partial}{\partial n_{k,c}}\left(\sum_a e_a\left(\frac{n_a}{n_{\mathcal{V}}}\right)\right) = \sum_{i\neq k, a\neq c} \frac{n_a n_{i,c}}{n_{\mathcal{V}}} - \sum_b \frac{e_b n_b}{n_{\mathcal{V}}^2} + \frac{e_c}{n_{\mathcal{V}}} \tag{23}$$

Expanding the third term in (21)

$$\frac{\partial}{\partial n_{k,c}}\left(\sum_a \gamma_a \left(\frac{n_a}{n_{\mathcal{V}}}\right)\right) = \frac{\partial}{\partial n_{k,c}}\left(\frac{n_c}{n_{\mathcal{V}}}\gamma_c\right) + \sum_{a\neq c}\frac{\partial}{\partial n_{k,c}}\left(\frac{n_a}{n_{\mathcal{V}}}\gamma_a\right)$$

$$= \frac{n_c}{n_{\mathcal{V}}}\underbrace{\frac{\partial}{\partial n_{k,c}}\gamma_c}_{=0} + \frac{\gamma_c}{n_{\mathcal{V}}}\underbrace{\frac{\partial}{\partial n_{k,c}}n_c}_{=1} - \gamma_c n_c \frac{\partial}{\partial n_{k,c}}\left(\frac{1}{n_{\mathcal{V}}}\right) + \sum_{a\neq c}\frac{\partial}{\partial n_{k,c}}\left(\frac{n_a}{n_{\mathcal{V}}}\gamma_a\right)$$

$$= \frac{\gamma_c}{n_{\mathcal{V}}} - \frac{\gamma_c n_c}{n_{\mathcal{V}}^2} + \sum_{a\neq c}\frac{\partial}{\partial n_{k,c}}\left(\frac{n_a}{n_{\mathcal{V}}}\gamma_a\right)$$

$$= \frac{\gamma_c}{n_{\mathcal{V}}} - \frac{\gamma_c n_c}{n_{\mathcal{V}}^2} + \sum_{a\neq c}\left(\frac{n_a}{n_{\mathcal{V}}}\underbrace{\left(\frac{\partial}{\partial n_{k,c}}\gamma_a\right)}_{=2n_{k,c}} + \frac{\gamma_a}{n_{\mathcal{V}}}\underbrace{\frac{\partial}{\partial n_{k,c}}n_a}_{=0} + \gamma_a n_a \frac{\partial}{\partial n_{k,c}}\frac{1}{n_{\mathcal{V}}}\right)$$

$$= \frac{\gamma_c}{n_{\mathcal{V}}} - \frac{\gamma_c n_c}{n_{\mathcal{V}}^2} + \sum_{a\neq c}\left(2\frac{n_a n_{k,c}}{n_{\mathcal{V}}} - \frac{\gamma_a n_a}{n_{\mathcal{V}}^2}\right)$$

$$= \frac{\gamma_c}{n_{\mathcal{V}}} + 2n_{k,c}\sum_{a\neq c}\frac{n_a}{n_{\mathcal{V}}} - \sum_b \frac{\gamma_b n_b}{n_{\mathcal{V}}^2}$$

$$\tag{24}$$

Putting it all together

$$\frac{\partial \phi}{\partial n_{k,c}} = \frac{(e_c + \gamma_c)}{n_{\mathcal{V}}} + \sum_{i\neq k, a\neq c}\frac{n_a n_{i,c}}{n_{\mathcal{V}}} + 2n_{k,c}\sum_{a\neq c}\frac{n_a}{n_{\mathcal{V}}} + \sum_{i\neq k, a\neq c} n_{i,a} - \sum_b \frac{e_b n_b}{n_{\mathcal{V}}^2} - \sum_b \frac{\gamma_b n_b}{n_{\mathcal{V}}^2} \tag{25}$$

Multiplying (25) by $n_{\mathcal{V}}$ we see it is non negative iff

$$\sum_b \frac{(e_b + \gamma_b)n_b}{n_{\mathcal{V}}} \leq e_c + \gamma_c + \sum_{a\neq c, i\neq k} n_a n_{i,c} + 2n_{k,c}\sum_{a\neq c} n_a + n_{\mathcal{V}}\sum_{a\neq c, i\neq k} n_{i,a} \tag{26}$$

Expanding LHS we get

$$\sum_b \frac{(e_b + \gamma_b)n_b}{n_{\mathcal{V}}} = \frac{(e_c + \gamma_c)n_c}{n_{\mathcal{V}}} + \sum_{b\neq c}\frac{(e_b + \gamma_b)n_b}{n_{\mathcal{V}}}$$

$$\leq e_c + \frac{\gamma_c n_c}{n_{\mathcal{V}}} + \sum_{b\neq c}\frac{(e_b + \gamma_b)n_b}{n_{\mathcal{V}}}$$

$$\leq e_c + \frac{\gamma_c n_c}{n_{\mathcal{V}}} + \sum_{b\neq c}\frac{n_b}{n_{\mathcal{V}}}\left(\sum_{i<j}\sum_{a\neq b} n_{i,a}.n_{j,a} + \sum_i \sum_{a\neq b} n_{i,a}^2\right)$$

$$\leq e_c + \frac{\gamma_c n_c}{n_{\mathcal{V}}} + \sum_{b\neq c}\frac{n_b}{n_{\mathcal{V}}}\left(\sum_{i<j} n_{i,c}.n_{j,c} + \sum_i n_{i,c}^2\right) + \sum_{b\neq c}\frac{n_b}{n_{\mathcal{V}}}\left(\sum_{i<j}\sum_{a\neq b,c} n_{i,a}.n_{j,a} + \sum_i \sum_{a\neq b,c} n_{i,a}^2\right)$$

$$\leq e_c + \underbrace{\sum_{b\neq c}\frac{n_b}{n_{\mathcal{V}}}\left(\sum_{i<j} n_{i,c}.n_{j,c} + \sum_i n_{i,c}^2\right)}_{\text{Ⓐ}} + \underbrace{\sum_{b\neq c}\frac{n_b}{n_{\mathcal{V}}}\left(\sum_{i<j}\sum_{a\neq b,c} n_{i,a}.n_{j,a} + \sum_i \sum_{a\neq b,c} n_{i,a}^2\right) + \frac{\gamma_c n_c}{n_{\mathcal{V}}}}_{\text{Ⓑ}}$$

$$\tag{27}$$

If $\{x_i\}_{i \geq 0}$ be a finite sequence of non-negative real numbers. Then for any $k$

$$\sum_{i<j} x_i x_j + \sum_i x_i^2 \leq \left(\sum_i x_i\right)\left(\sum_{i \neq k} x_i\right) + x_k^2 \tag{28}$$

Using (28) and expanding Ⓐ we have

$$\begin{aligned}
&\sum_{b \neq c} \frac{n_b}{n_{\mathcal{V}}}\left(\sum_{i<j} n_{i,c}.n_{j,c} + \sum_i n_{i,c}^2\right) \\
&\leq \sum_{b \neq c} \frac{n_b}{n_{\mathcal{V}}}\left(\left(\sum_i n_{i,c}\right)\left(\sum_{i \neq k} n_{i,c}\right) + n_{k,c}^2\right) \\
&\leq \sum_{b \neq c} \frac{n_b}{n_{\mathcal{V}}}\left(n_c\left(\sum_{i \neq k} n_{i,c}\right) + n_{k,c}^2\right) \\
&\leq \sum_{b \neq c} \frac{n_b n_c}{n_{\mathcal{V}}}\left(\sum_{i \neq k} n_{i,c}\right) + \sum_{b \neq c} \frac{n_b}{n_{\mathcal{V}}} n_{k,c}^2 \\
&\leq \sum_{b \neq c} n_b\left(\sum_{i \neq k} n_{i,c}\right) + \sum_{b \neq c} \frac{n_b n_c}{n_{\mathcal{V}}} n_{k,c} \\
&\leq \sum_{a \neq c, i \neq k} n_a n_{i,c} + \sum_{b \neq c} n_b n_{k,c} \\
&\leq \sum_{a \neq c, i \neq k} n_a n_{i,c} + 2 n_{k,c} \sum_{a \neq c} n_a
\end{aligned} \tag{29}$$

Using (28) and expanding Ⓑ we have

$$\begin{aligned}
&\sum_{b \neq c} \frac{n_b}{n_{\mathcal{V}}}\left(\sum_{i<j} \sum_{a \neq b,c} n_{i,a}.n_{j,a} + \sum_i \sum_{a \neq b,c} n_{i,a}^2\right) + \frac{\gamma_c n_c}{n_{\mathcal{V}}} \\
&\leq \sum_{b \neq c} \frac{n_b}{n_{\mathcal{V}}}\left(\sum_{a \neq b,c}\left(\sum_{i \neq k} n_{i,a}\right) n_a + \sum_{a \neq b,c} n_{k,a}^2\right) + \frac{\gamma_c n_c}{n_{\mathcal{V}}} \\
&\leq \sum_{b \neq c} \frac{n_b}{n_{\mathcal{V}}} n_{\mathcal{V}} \sum_{a \neq c} \sum_{i \neq k} n_{i,a} + \sum_{b \neq c} \frac{n_b}{n_{\mathcal{V}}} \sum_{a \neq b,c} n_{k,a}^2 + \frac{\gamma_c n_c}{n_{\mathcal{V}}} \\
&\leq \sum_{b \neq c} n_b \sum_{a \neq c, i \neq k} n_{i,a} + \sum_{b \neq c} \frac{n_b}{n_{\mathcal{V}}} \sum_{a \neq c} n_{k,a}^2 + \frac{\gamma_c n_c}{n_{\mathcal{V}}} \\
&\leq n_{\mathcal{V}} \sum_{a \neq c, i \neq k} n_{i,a} + \sum_{b \neq c} \frac{n_b}{n_{\mathcal{V}}} \gamma_c + \frac{\gamma_c n_c}{n_{\mathcal{V}}} \\
&\leq n_{\mathcal{V}} \sum_{a \neq c, i \neq k} n_{i,a} + \gamma_c\left(\sum_{b \neq c} \frac{n_b}{n_{\mathcal{V}}} + \frac{n_c}{n_{\mathcal{V}}}\right) \\
&\leq n_{\mathcal{V}} \sum_{a \neq c, i \neq k} n_{i,a} + \gamma_c
\end{aligned} \tag{30}$$

Combining (29) and (30)

$$\sum_b \frac{(e_b + \gamma_b)n_b}{n_{\mathcal{V}}} \leq e_c + \sum_{a \neq c, i \neq k} n_a n_{i,c} + 2 n_{k,c} \sum_{a \neq c} n_a + n_{\mathcal{V}} \sum_{a \neq c, i \neq k} n_{i,a} + \gamma_c \tag{31}$$

Hence the inequality $\frac{\partial \phi}{\partial n_{k,c}} \geq 0$ holds. For non-uniform prior, the proofs from Lemma 6 in Golovin et al. [17] carry over.

<div align="right">□</div>

## Appendix B  Proof of Lemma 2

**Lemma 3.** *The expression $f_{\mathrm{DRD}}(\mathbf{x}_{\mathcal{A}})$ is strongly adaptive monotone and adaptive submodular.*

*Proof.* We adapt the proof from Lemma 1 in Chen et al. [5]. $f_{\mathrm{DRD}}(\mathbf{x}_{\mathcal{A}})$ can be shown to be strongly adaptive monotone from Chen et al. [5] by showing each individual $f_{\mathrm{EC}}^i(\mathbf{x}_{\mathcal{A}})$ is strongly adaptive monotone.

To proof adaptive submodularity, we must show that for all $\mathbf{x}_{\mathcal{A}} < \mathbf{x}_{\mathcal{B}}$ and $t \in \mathcal{T}$, we have $\Delta_{f_{\mathrm{DRD}}}(t \mid \mathbf{x}_{\mathcal{A}}) \geq \Delta_{f_{\mathrm{DRD}}}(t \mid \mathbf{x}_{\mathcal{B}})$. We first show this for two problems in the noisy OR formulation.

As shown in (7) in Chen et al. [5], we have
$$\Delta_{f_{\mathrm{DRD}}}(t \mid \mathbf{x}_{\mathcal{A}}) = (1 - f_{\mathrm{EC}}^1(\mathbf{x}_{\mathcal{A}}))\mathbb{E}_{x_t}\left[\delta_2(x_t|\mathbf{x}_{\mathcal{A}})|\mathbf{x}_{\mathcal{A}}\right] + \mathbb{E}_{x_t}\left[(1 - f_{\mathrm{EC}}^2(\mathbf{x}_{\mathcal{A}\cup\{t\}}))\delta_1(x_t|\mathbf{x}_{\mathcal{A}})|\mathbf{x}_{\mathcal{A}}\right] \tag{32}$$

The first term satisfies
$$(1 - f_{\mathrm{EC}}^1(\mathbf{x}_{\mathcal{A}}))\mathbb{E}_{x_t}\left[\delta_2(x_t|\mathbf{x}_{\mathcal{A}})|\mathbf{x}_{\mathcal{A}}\right] \geq (1 - f_{\mathrm{EC}}^1(\mathbf{x}_{\mathcal{B}}))\mathbb{E}_{x_t}\left[\delta_2(x_t|\mathbf{x}_{\mathcal{B}})|\mathbf{x}_{\mathcal{B}}\right] \tag{33}$$

Let the second term be $\lambda(\mathbf{n})$ and denote $h(\mathbf{n}) = (1 - f_{\mathrm{EC}}^2(\mathbf{x}_{\mathcal{A}\cup\{t\}}))$. We will show $\frac{\partial \lambda(\mathbf{n})}{\partial n_{k,c}} \geq 0$ for all $n_{k,c}$.

$$\lambda(\mathbf{n}) = \mathbb{E}_{x_t}\left[h(\mathbf{n})\delta_1(x_t|\mathbf{x}_{\mathcal{A}})|\mathbf{x}_{\mathcal{A}}\right] \tag{34}$$

The partial derivative $\frac{\partial \lambda(\mathbf{n})}{\partial n_{k,c}}$ can be expressed as

$$\frac{\partial \lambda(\mathbf{n})}{\partial n_{k,c}} = \sum_a \frac{\partial h(\mathbf{n})}{\partial n_{k,c}}\left(\frac{n_a}{n_{\mathcal{V}}}p + \frac{n_a}{n_{\mathcal{V}}}e_a + \frac{n_a}{n_{\mathcal{V}}}\gamma_a\right) + \sum_a h(\mathbf{n})\frac{\partial}{\partial n_{k,c}}\left(\frac{n_a}{n_{\mathcal{V}}}p + \frac{n_a}{n_{\mathcal{V}}}e_a + \frac{n_a}{n_{\mathcal{V}}}\gamma_a\right) \tag{35}$$

Since $\frac{\partial h(\mathbf{n})}{\partial n_{k,c}} \geq 0$, the first term is $\geq 0$. Expanding the second term, we have

$$\sum_a h(\mathbf{n})\frac{\partial}{\partial n_{k,c}}\left(\frac{n_a}{n_{\mathcal{V}}}p + \frac{n_a}{n_{\mathcal{V}}}e_a + \frac{n_a}{n_{\mathcal{V}}}\gamma_a\right)$$
$$= h(\mathbf{n})\underbrace{\frac{\partial}{\partial n_{k,c}}\left(\frac{n_c}{n_{\mathcal{V}}}p + \frac{n_c}{n_{\mathcal{V}}}e_c + \frac{n_c}{n_{\mathcal{V}}}\gamma_c\right)}_{\text{\textcircled{A}}} + \sum_{a\neq c} h(\mathbf{n})\underbrace{\frac{\partial}{\partial n_{k,c}}\left(\frac{n_a}{n_{\mathcal{V}}}p + \frac{n_a}{n_{\mathcal{V}}}e_a + \frac{n_a}{n_{\mathcal{V}}}\gamma_a\right)}_{\text{\textcircled{B}}} \tag{36}$$

Expanding first term in \textcircled{A}
$$\frac{\partial}{\partial n_{k,c}}\left(\frac{n_a}{n_{\mathcal{V}}}p\right) = \frac{n_c}{n_{\mathcal{V}}}\underbrace{\frac{\partial}{\partial n_{k,c}}p}_{=\sum_{i\neq k,b\neq c} n_{i,b}} + \frac{p}{n_{\mathcal{V}}}\underbrace{\frac{\partial}{\partial n_{k,c}}n_c}_{=1} + pn_c\frac{\partial}{\partial n_{k,c}}\left(\frac{1}{n_{\mathcal{V}}}\right)$$
$$= \frac{n_c}{n_{\mathcal{V}}}\sum_{i\neq k,b\neq c} n_{i,b} + \frac{p}{n_{\mathcal{V}}}\left(1 - \frac{n_c}{n_{\mathcal{V}}}\right) \tag{37}$$

Expanding second term in \textcircled{A}
$$\frac{\partial}{\partial n_{k,c}}\left(\frac{n_c}{n_{\mathcal{V}}}e_c\right) = \frac{n_c}{n_{\mathcal{V}}}\underbrace{\frac{\partial}{\partial n_{k,c}}e_c}_{=0} + \frac{e_c}{n_{\mathcal{V}}}\underbrace{\frac{\partial}{\partial n_{k,c}}n_c}_{=1} + e_c n_c\frac{\partial}{\partial n_{k,c}}\left(\frac{1}{n_{\mathcal{V}}}\right)$$
$$= \frac{e_c}{n_{\mathcal{V}}}\left(1 - \frac{n_c}{n_{\mathcal{V}}}\right) \tag{38}$$

<div align="center">24</div>

Expanding second term in $\text{(A)}$

$$\frac{\partial}{\partial n_{k,c}}\left(\frac{n_c}{n_\mathcal{V}}\gamma_c\right) = \frac{n_c}{n_\mathcal{V}}\underbrace{\frac{\partial}{\partial n_{k,c}}\gamma_c}_{=0} + \frac{\gamma_c}{n_\mathcal{V}}\underbrace{\frac{\partial}{\partial n_{k,c}}n_c}_{=1} + \gamma_c n_c\frac{\partial}{\partial n_{k,c}}\left(\frac{1}{n_\mathcal{V}}\right) \tag{39}$$

$$= \frac{\gamma_c}{n_\mathcal{V}}\left(1 - \frac{n_c}{n_\mathcal{V}}\right)$$

Putting things together $\text{(A)}$ evaluates to

$$\frac{n_c}{n_\mathcal{V}}\sum_{i\neq k,b\neq c}n_{i,b} + \frac{p}{n_\mathcal{V}}\left(1 - \frac{n_c}{n_\mathcal{V}}\right) + \frac{e_c}{n_\mathcal{V}}\left(1 - \frac{n_c}{n_\mathcal{V}}\right) + \frac{\gamma_c}{n_\mathcal{V}}\left(1 - \frac{n_c}{n_\mathcal{V}}\right) \tag{40}$$

Now the first term in $\text{(B)}$ evaluates to

$$\frac{\partial}{\partial n_{k,c}}\left(\frac{n_a}{n_\mathcal{V}}p\right) = \frac{n_a}{n_\mathcal{V}}\underbrace{\frac{\partial}{\partial n_{k,c}}p}_{=\sum_{i\neq k,b\neq c}n_{i,b}} + \frac{p}{n_\mathcal{V}}\underbrace{\frac{\partial}{\partial n_{k,c}}n_a}_{=1} + pn_a\frac{\partial}{\partial n_{k,c}}\left(\frac{1}{n_\mathcal{V}}\right) \tag{41}$$

$$= \frac{n_a}{n_\mathcal{V}}\sum_{i\neq k,b\neq c}n_{i,b} - \frac{pn_a}{n_\mathcal{V}^2}$$

The second term in $\text{(B)}$ evaluates to

$$\frac{\partial}{\partial n_{k,c}}\left(\frac{n_a}{n_\mathcal{V}}e_a\right) = \frac{n_a}{n_\mathcal{V}}\underbrace{\frac{\partial}{\partial n_{k,c}}e_a}_{=\sum_{i\neq k}n_{i,c}} + \frac{e_a}{n_\mathcal{V}}\underbrace{\frac{\partial}{\partial n_{k,c}}n_a}_{=0} + e_a n_a\frac{\partial}{\partial n_{k,c}}\left(\frac{1}{n_\mathcal{V}}\right) \tag{42}$$

$$= \frac{n_a}{n_\mathcal{V}}\left(\sum_{i\neq k}n_{i,c} - \frac{e_a}{n_\mathcal{V}}\right)$$

The third term in $\text{(B)}$ evaluates to

$$\frac{\partial}{\partial n_{k,c}}\left(\frac{n_a}{n_\mathcal{V}}\gamma_a\right) = \frac{n_a}{n_\mathcal{V}}\underbrace{\frac{\partial}{\partial n_{k,c}}\gamma_a}_{=2n_{k,c}} + \frac{\gamma_a}{n_\mathcal{V}}\underbrace{\frac{\partial}{\partial n_{k,c}}n_a}_{=0} + \gamma_a n_a\frac{\partial}{\partial n_{k,c}}\left(\frac{1}{n_\mathcal{V}}\right) \tag{43}$$

$$= \frac{n_a}{n_\mathcal{V}}\left(2n_{k,c} - \frac{\gamma_a}{n_\mathcal{V}}\right)$$

Combining $\text{(B)}$

$$\frac{n_a}{n_\mathcal{V}}\sum_{i\neq k,b\neq c}n_{i,b} - \frac{pn_a}{n_\mathcal{V}^2} + \frac{n_a}{n_\mathcal{V}}\left(\sum_{i\neq k}n_{i,c} - \frac{e_a}{n_\mathcal{V}}\right) + \frac{n_a}{n_\mathcal{V}}\left(\sum_{i\neq k}2n_{k,c} - \frac{\gamma_a}{n_\mathcal{V}}\right)$$

$$\frac{n_a}{n_\mathcal{V}}\left(\sum_{i\neq k,b\neq c}n_{i,b} + \sum_{i\neq k}n_{i,c} + 2n_{k,c} - \frac{1}{n_\mathcal{V}}\left(p + e_a + \gamma_a\right)\right) \tag{44}$$

Combining Ⓐ and Ⓑ, (36) can be evaluated as

$$
h(\mathbf{n}) \left( \underbrace{\frac{n_c}{n_\mathcal{V}} \sum_{i \neq k, b \neq c} n_{i,b}}_{\geq 0} + \underbrace{\frac{p}{n_\mathcal{V}} \left(1 - \frac{n_c}{n_\mathcal{V}}\right)}_{\geq 0} + \underbrace{\frac{e_c}{n_\mathcal{V}} \left(1 - \frac{n_c}{n_\mathcal{V}}\right)}_{\geq 0} + \frac{\gamma_c}{n_\mathcal{V}} \left(1 - \frac{n_c}{n_\mathcal{V}}\right) \right) +
$$

$$
\sum_{a \neq c} h(\mathbf{n}) \frac{n_a}{n_\mathcal{V}} \left( \sum_{i \neq k, b \neq c} n_{i,b} + \sum_{i \neq k} n_{i,c} + 2 n_{k,c} - \frac{1}{n_\mathcal{V}} \left(p + e_a + \gamma_a\right) \right)
$$

$$
\geq h(\mathbf{n}) \frac{\gamma_c}{n_\mathcal{V}} \left(1 - \frac{n_c}{n_\mathcal{V}}\right) + \sum_{a \neq c} h(\mathbf{n}) \frac{n_a}{n_\mathcal{V}} \left( \sum_{i \neq k, b} n_{i,b} + 2 n_{k,c} - \frac{1}{n_\mathcal{V}} \left(p + e_a + \gamma_a\right) \right)
$$

$$
\geq h(\mathbf{n}) \gamma_c \left(1 - \frac{n_c}{n_\mathcal{V}}\right) + \sum_{a \neq c} h(\mathbf{n}) \frac{n_a}{n_\mathcal{V}} \left( n_\mathcal{V} \left( \sum_{i \neq k, b} n_{i,b} + 2 n_{k,c} \right) - \underbrace{\left(p + e_a + \gamma_a\right)}_{\text{Ⓒ}} \right)
$$
(45)

Expanding Ⓒ we have

$$
(p + e_a + \gamma_a) \leq \sum_{i<j} \sum_{b \neq d} n_{i,b} n_{j,d} + \sum_{i<j} \sum_{b \neq a,c} n_{i,b} n_{j,b} + \sum_{i} \sum_{b \neq a,c} n_{i,b}^2
$$

$$
\leq \sum_{i<j} \sum_{b \neq d} n_{i,b} n_{j,d} + \sum_{i<j} \sum_{b \neq d} n_{i,b} n_{j,b} + \sum_{i \neq k} \sum_{b \neq a,c} n_{i,b}^2 + \sum_{b \neq a,c} n_{k,b}^2
$$

$$
\leq \left( \sum_{i,d} n_{i,d} \right) \left( \sum_{j \neq k} \sum_{b} n_{j,b} \right) + \sum_{b \neq a,c} n_{k,b}^2
$$
(46)

$$
\leq n_\mathcal{V} \left( \sum_{j \neq k} \sum_{b} n_{j,b} \right) + \gamma_c
$$

Substituting in (45) we have

$$
\geq h(\mathbf{n}) \gamma_c \left(1 - \frac{n_c}{n_\mathcal{V}}\right) + \sum_{a \neq c} h(\mathbf{n}) \frac{n_a}{n_\mathcal{V}} \left( n_\mathcal{V} \left( \sum_{i \neq k, b} n_{i,b} + 2 n_{k,c} \right) - \left(p + e_a + \gamma_a\right) \right)
$$

$$
\geq h(\mathbf{n}) \gamma_c \left(1 - \frac{n_c}{n_\mathcal{V}}\right) + \sum_{a \neq c} h(\mathbf{n}) \frac{n_a}{n_\mathcal{V}} \left( n_\mathcal{V} \left( \sum_{i \neq k, b} n_{i,b} + 2 n_{k,c} \right) - n_\mathcal{V} \left( \sum_{j \neq k} \sum_{b} n_{j,b} \right) + \gamma_c \right)
$$

$$
\geq h(\mathbf{n}) \gamma_c \left(1 - \frac{n_c}{n_\mathcal{V}}\right) + \sum_{a \neq c} h(\mathbf{n}) \frac{n_a}{n_\mathcal{V}} \gamma_c
$$

$$
\geq 0
$$
(47)

Hence we have proved $\frac{\partial \lambda(\mathbf{n})}{\partial n_{k,c}} \geq 0$ for all $n_{k,c}$. This implies adaptive submodularity is proved for 2 regions. For more than 2, we apply the recursive technique in Lemma 1 in Chen et al. [5].

□

# Appendix C    Proof of Theorem 1

**Theorem.** *Let $m$ be the number of regions, $p_{\min}^h$ the minimum prior probability of any hypothesis, $\pi_{DRD}$ be the greedy policy and $\pi^*$ with the optimal policy. Then $c(\pi_{DRD}) \leq c(\pi^*)(2m \log \frac{1}{p_{\min}^h} + 1)$.*

*Proof.* This is a straightforward application of Theorem 2 in Chen et al. [5].    □

# Appendix D   Proof of Theorem 2

**Theorem.** *A policy that greedily latches to a region according the the posterior conditioned on the region outcomes has a near-optimality guarantee of 4 w.r.t the optimal region evaluation sequence.*

*Proof.* We establish an equivalence to the problem of greedy search on a binary vector as described in Dor [13].

**Problem 1.** *Consider the $n$-dimensional binary space with some (general) probability distribution. Suppose that a random vector is sampled from this space and it is initially unseen. A search algorithm on such a vector is a procedure inspecting one coordinate at a time in a pre-determined order. It terminates when a 1-coordinate is found or when all coordinates were tested and found to be 0. A greedy search is one that goes at each stage to the next coordinate most likely to be 1, taking into account the findings of the previous examinations and the distribution. Can we bound the performance of the greedy search with a search optimal in expectation?*

Dor [13] proves the following

**Theorem 5.** *The expectation of the greedy algorithm (denoted $E_1$) is always less than $4$ times the expectation of the optimal algorithm (denoted $E_2$)*

If we imagine each region $\mathcal{R}_i$ to be a coordinate, then an algorithm that greedily selects regions to evaluate based on the outcomes of previous region check has a bounded sub-optimality. We note that MAXPROBREG uses a more accurate posterior as it has access to the individual results of edge evaluation. Hence it is expected to do better than the greedy algorithm that only conditions on the outcome of the region evaluation.

$\square$

# Appendix E   Proof of Theorem 3

**Theorem.** *Let $p_{\min} = \min_i P(\mathcal{R}_i)$, $p_{\min}^h = \min_{h \in \mathcal{H}} P(h)$ and $l = \max_i |\mathcal{R}_i|$. The policy using (15) has a suboptimality of $\alpha \left( 2m \log \left( \frac{1}{p_{\min}^h} \right) + 1 \right)$ where $\alpha \leq \left( 1 - \max \left( (1 - p_{\min})^2, p_{\min}^{\frac{2}{l}} \right) \right)^{-1}$.*

*Proof.* We start of by defining policies that do not greedily maximize $f_{\mathrm{DRD}}(\mathbf{x}_{\mathcal{A}})$

**Definition 1.** *Let an $\alpha$-approximate greedy policy be one that selects a test $t'$ that satisfies the following criteria*

$$\Delta_{f_{\mathrm{DRD}}} \left( t' \mid \mathbf{x}_{\mathcal{A}} \right) \geq \frac{1}{\alpha} \max_t \Delta_{f_{\mathrm{DRD}}} \left( t \mid \mathbf{x}_{\mathcal{A}} \right)$$

We examine the scenarios where cost is uniform $c(t) = 1$ for the sake of simplicity - the proof can be easily extended to non-uniform setting. We refer to the policy using (15) as a test constraint as a MAXPROBREG policy.

The marginal gain is evaluated as follows

$$
\begin{aligned}
\Delta_{f_{\mathrm{DRD}}} \left( t \mid \mathbf{x}_{\mathcal{A}} \right) = \mathbb{E}_{x_t} &\left[ \prod_{r=1}^{m} \left( 1 - \prod_{i \in (\mathcal{R}_r \cap \mathcal{A})} \mathbb{I}(X_i = 1) \prod_{j \in (\mathcal{R}_r \setminus \mathcal{A})} \theta_j \right) \right. \\
&\left. - \left( \prod_{r=1}^{m} \left( 1 - \prod_{i \in (\mathcal{R}_r \cap \mathcal{A} \cup t)} \mathbb{I}(X_i = 1) \prod_{j \in (\mathcal{R}_r \setminus \mathcal{A} \cup t)} \theta_j \right) \right) (\theta_t^{x_t} (1 - \theta_t)^{1 - x_t})^{2 \sum\limits_{k=1}^{m} \mathbb{I}(t \in \mathcal{R}_k)} \right]
\end{aligned}
$$
(48)

We will now bound $\alpha$, the ratio of marginal gain of the unconstrained greedy policy and MAXPRO-BREG.

Figure 14: The scenario where sub-optimality bound is maximized

$$\alpha \leq \frac{\max\limits_{t \in \mathcal{T}_{\text{cand}}} \Delta_{f_{\text{DRD}}}(t \mid \mathbf{x}_{\mathcal{A}})}{\min\limits_{t \in \mathcal{T}_{\text{maxP}}} \Delta_{f_{\text{DRD}}}(t \mid \mathbf{x}_{\mathcal{A}})} \tag{49}$$

The numerator and denominator contain $\prod\limits_{r=1}^{m} \left( 1 - \prod\limits_{i \in (\mathcal{R}_r \cap \mathcal{A})} \mathbb{I}(X_i = 1) \prod\limits_{j \in (\mathcal{R}_r \setminus \mathcal{A})} \theta_j \right)$, the posterior probabilities of regions not being valid. Hence we normalize by dividing this term and expressing $\alpha$ in terms of a residual function $\rho(t|\mathbf{x}_{\mathcal{A}})$

$$\alpha \leq \frac{1 - \min\limits_{t \in \mathcal{T}_{\text{cand}}} \rho(t|\mathbf{x}_{\mathcal{A}})}{1 - \max\limits_{t \in \mathcal{T}_{\text{maxP}}} \rho(t|\mathbf{x}_{\mathcal{A}})} \tag{50}$$

where the residual function $\rho(t|\mathbf{x}_{\mathcal{A}})$ is

$$\rho(t|\mathbf{x}_{\mathcal{A}}) = \frac{\mathbb{E}_{x_t} \left[ \left( \prod\limits_{r=1}^{m} \left( 1 - \prod\limits_{i \in (\mathcal{R}_r \cap \mathcal{A} \cup t)} \mathbb{I}(X_i = 1) \prod\limits_{j \in (\mathcal{R}_r \setminus \mathcal{A} \cup t)} \theta_j \right) \right) \left( \theta_t^{x_t} (1 - \theta_t)^{1-x_t} \right)^{2 \sum\limits_{k=1}^{m} \mathbb{I}(t \in \mathcal{R}_k)} \right]}{\prod\limits_{r=1}^{m} \left( 1 - \prod\limits_{i \in (\mathcal{R}_r \cap \mathcal{A})} \mathbb{I}(X_i = 1) \prod\limits_{j \in (\mathcal{R}_r \setminus \mathcal{A})} \theta_j \right)} \tag{51}$$

We will now claim that the bound is maximized in the scenario shown in Fig. 14. The most likely region, $R_1$, is an isolated path which contains no tests in common with other regions. Let the probability of this region be $p_{\min}$ - this is the smallest probability that can be assigned to it. All other $m - 1$ regions (of lower probabilty) share a common test $a$ of probability $\theta_a$. The remaining tests in these regions have probability 1. Note that $\theta_a \leq p_{\min}$. In this scenario, the greedy policy will select the common test while the MAXPROBREG will select a test from the most probably region.

We will now show that this scenario allows us to realize the upper bound 1 for the numerator in (50). In other words, we will show that the residual function $\rho(a|\mathbf{x}_{\mathcal{A}}) \ll 1$ in our scenario.

$$\rho(a|\mathbf{x}_{\mathcal{A}}) \leq \frac{\theta_a p_{\min} \prod\limits_{r=1}^{m-1} (1-1)\theta_a^{2(m-1)} \; + \; (1-\theta_a)p_{\min} \prod\limits_{r=1}^{m-1} (1-0)(1-\theta_a)^{2(m-1)}}{p_{\min} \prod\limits_{r=1}^{m-1} (1-\theta_a)}$$

$$\leq \frac{(1-\theta_a)^{2m-1}}{\prod\limits_{r=1}^{m-1} (1-\theta_a)} \tag{52}$$

$$\leq (1-\theta_a)^m$$

We can drive $\rho(a|\mathbf{x}_{\mathcal{A}}) \ll 1$ by setting $m$ arbitrarily high.

We now show that scenario also allows us to bound the denominator in (50) by maximizing $\max\limits_{t \in \mathcal{R}_1} \rho(t|\mathbf{x}_{\mathcal{A}})$. We first note that by selecting a test that belongs only to one region, the residual is maximized. We have to figure out how large the residual can be. Let $\tau = \arg\max\limits_{t \in \mathcal{R}_1} \rho(t|\mathbf{x}_{\mathcal{A}})$ be the most probable test with probability $\theta_\tau$. Let $\beta = \prod\limits_{t \in \mathcal{R}_1, t \neq \tau} \theta_t$ be the lumped probability of all other

tests. Note that $\theta_\tau \beta = p_{\min}$. The residual can be expressed as

$$\rho(\tau|\mathbf{x}_\mathcal{A}) \leq \frac{\theta_\tau \prod_{r=1}^{m-1}(1-\theta_a)(1-\beta)\theta_\tau^2 \; + \; (1-\theta_\tau)\prod_{r=1}^{m-1}(1-\theta_a)(1-\theta_\tau)^2}{\prod_{r=1}^{m-1}(1-\theta_a)(1-\theta_\tau\beta)}$$

$$\leq \frac{\theta_\tau^3(1-\beta)+(1-\theta_\tau)^3}{(1-\theta_\tau\beta)}$$

$$\leq \frac{\theta_\tau^3(1-\beta)+\theta_\tau^2-\theta_\tau^2+(1-\theta_\tau)^3}{(1-\theta_\tau\beta)} \tag{53}$$

$$\leq \frac{\theta_\tau^2(1-\theta_\tau\beta)-\theta_\tau^2(1-\theta_\tau)+(1-\theta_\tau)^3}{(1-\theta_\tau\beta)}$$

$$\leq \theta_\tau^2 - \frac{(1-\theta_\tau)(2\theta_\tau-1)}{(1-p_{\min})}$$

This bound is concave and achieves maxima on the two extrema. In the first case, we assume $\beta = 0$, $\theta_\tau = p_{\min}$. This leads to

$$\rho(\tau|\mathbf{x}_\mathcal{A}) \leq p_{\min}^2 - \frac{(1-p_{\min})(2p_{\min}-1)}{(1-p_{\min})}$$

$$\leq p_{\min}^2 - (2p_{\min}-1) \tag{54}$$

$$(1-p_{\min})^2$$

In the second case, we assume $\beta = \theta_\tau$. Lete $l$ be the maximum test in any region. Then $\theta_\tau = p_{\min}^{\frac{1}{l}}$. This leads to

$$\rho(\tau|\mathbf{x}_\mathcal{A}) \leq \theta_\tau^2 - \frac{(1-\theta_\tau)(2\theta_\tau-1)}{(1-p_{\min})}$$

$$\leq \theta_\tau^2 \tag{55}$$

$$\leq p_{\min}^{\frac{2}{l}}$$

Combining these we have

$$\rho(\tau|\mathbf{x}_\mathcal{A}) \leq \max\left((1-p_{\min})^2, p_{\min}^{\frac{2}{l}}\right) \tag{56}$$

Substituting (56) in (50) we have

$$\alpha \leq \frac{1}{1-\max\left((1-p_{\min})^2, p_{\min}^{\frac{2}{l}}\right)} \tag{57}$$

We now use Theorem 11 in Golovin and Krause [16] to state that an $\alpha$-approximate greedy policy $\pi$ optimizing $f_{\mathrm{DRD}}(\mathbf{x}_\mathcal{A})$ enjoys the following guarantee

$$c(\pi) \leq \alpha c(\pi^*)\left(2m\log\left(\frac{1}{p_{\min}^h}\right)+1\right) \tag{58}$$

$\square$

# Appendix F    Proof of Theorem 4

**Theorem.** SETCOVER *is a near-optimal policy for checking all regions.*

We present a refined version of the theorem that we will prove.
**Theorem.** *Let $\pi$ be the* SETCOVER *policy, which is a partial mapping from observation vector $\mathbf{x}_\mathcal{A}$ to tests, such that it terminates only when all regions $\mathcal{R}_i$ are either completely evaluated or invalidated. Let the expected cost of such a policy be $c(\pi)$. Let $\pi^*$ be the optimal policy for checking all regions. Let $n = |\mathcal{T}|$ be the number of tests.* SETCOVER *enjoys the following guarantee*

$$c(\pi) \leq c(\pi^*)(\log(n)+1)$$

*Proof.* We will prove this by drawing an equivalence of the problem to a special case of *stochastic set coverage* with non-uniform costs, showing SETCOVER greedily solves this problem, and using a guarantee for a greedy policy as presented in Golovin and Krause [16].

The stochastic set coverage problem is as follows - there is a ground set of elements $U$, and items $E$ such that item $e$ is associated with a distribution over subsets of $U$. When an item is selected, a set is sampled from its distribution. The problem is to adaptively select items until all elements of $U$ are covered by sampled sets, while minimizing the expected number of items selected. Here we consider the case where a cost is associated with each item.

We now show that the problem of selecting tests to invalidate other tests is equivalent to stochastic set coverage. The ground set is the set of all tests $\mathcal{T}$. The item set has a one to one correspondence with the test set $\mathcal{T}$. Let $\hat{f}(\mathbf{x}_{\mathcal{A}})$ be the utility function measuring coverage of tests given selected tests and outcomes $\mathbf{x}_{\mathcal{A}}$. This is defined as

$$\hat{f}(\mathbf{x}_{\mathcal{A}}) = \left| \mathcal{A} \cup \left\{ \bigcup_{i=1}^{m} \{ \mathcal{R}_i \mid P(\mathcal{R}_i | \mathbf{x}_{\mathcal{A}}) = 0 \} \right\} \right| \tag{59}$$

The expected gain in utility when selecting a test $t$ is as follows - with probability $\theta_t$ if a $t$ outcome is true, only $t$ is covered. With probability $1 - \theta_t$, if a test outcome is false, tests that belong to regions being invalidated are covered. This can be expressed formally as follows. Given $t \notin \left\{ \bigcup_{i=1}^{m} \{ \mathcal{R}_i \mid P(\mathcal{R}_i | \mathbf{x}_{\mathcal{A}}) = 0 \} \right\}$, the expected gain $\Delta_{\hat{f}} (t \mid \mathbf{x}_{\mathcal{A}})$ is

$$\Delta_{\hat{f}} (t \mid \mathbf{x}_{\mathcal{A}}) = \mathbb{E}_{x_t} \left[ \hat{f}(\mathbf{x}_{\mathcal{A} \cup \{t\}}) - \hat{f}(\mathbf{x}_{\mathcal{A}}) \right]$$

$$= \mathbb{E}_{x_t} \left[ \left| \mathcal{A} \cup \{t\} \cup \left\{ \bigcup_{i=1}^{m} \{ \mathcal{R}_i \mid P(\mathcal{R}_i | \mathbf{x}_{\mathcal{A} \cup \{t\}}) = 0 \} \right\} \right| - \left| \mathcal{A} \cup \left\{ \bigcup_{i=1}^{m} \{ \mathcal{R}_i \mid P(\mathcal{R}_i | \mathbf{x}_{\mathcal{A}}) = 0 \} \right\} \right| \right]$$

$$= P(X_t = 1) \times 1 +$$

$$P(X_t = 0) \times \left( 1 + \left| \left\{ \bigcup_{i=1}^{m} \{ \mathcal{R}_i \mid P(\mathcal{R}_i | \mathbf{x}_{\mathcal{A}}) > 0 \} - \bigcup_{j=1}^{m} \{ \mathcal{R}_j \mid P(\mathcal{R}_j | \mathbf{x}_{\mathcal{A}}, X_t = 0) > 0 \} \right\} \setminus \{ \mathcal{A} \cup \{t\} \} \right| \right)$$

$$= 1 + (1 - \theta_t) \left| \left\{ \bigcup_{i=1}^{m} \{ \mathcal{R}_i \mid P(\mathcal{R}_i | \mathbf{x}_{\mathcal{A}}) > 0 \} - \bigcup_{j=1}^{m} \{ \mathcal{R}_j \mid P(\mathcal{R}_j | \mathbf{x}_{\mathcal{A}}, X_t = 0) > 0 \} \right\} \setminus \{ \mathcal{A} \cup \{t\} \} \right| \tag{60}$$

SETCOVER is an adaptive greedy policy with respect to $\Delta_{\hat{f}} (t \mid \mathbf{x}_{\mathcal{A}})$ as shown

$$t^* \in \arg\max_{t \in \mathcal{T}} \Delta_{\hat{f}} (t \mid \mathbf{x}_{\mathcal{A}})$$

$$\in \arg\max_{t \in \mathcal{T}} (1 - \theta_t) \left| \left\{ \bigcup_{i=1}^{m} \{ \mathcal{R}_i \mid P(\mathcal{R}_i | \mathbf{x}_{\mathcal{A}}) > 0 \} - \bigcup_{j=1}^{m} \{ \mathcal{R}_j \mid P(\mathcal{R}_j | \mathbf{x}_{\mathcal{A}}, X_t = 0) > 0 \} \right\} \setminus \{ \mathcal{A} \cup \{t\} \} \right| \tag{61}$$

Note that $\hat{f}(\mathbf{x}_{\mathcal{T}}) = n$ is the maximum value the utility can attain. Let $\pi^*$ be the optimal policy. Since $\hat{f}(\mathbf{x}_{\mathcal{A}})$ is a strong adaptive monotone submodular function, we use Theorem 15 in Golovin and Krause [16] to state the following guarantee

$$c(\pi) \leq c(\pi^*)(\log(n) + 1)$$

$\square$

## Appendix G   Datasets with large disparity in region sizes

In this section, we investigate scenarios where there is a large disparity in region sizes. We will show that in such scenarios, MAXPROBREG has an arbitrarily poor performance. We will also show that unconstrained BISECT vastly outperforms all other algorithms on such problems.

We first examine the scenario as shown in Fig. 15(a). There are two regions $\mathcal{R}_1$ and $\mathcal{R}_2$. $\mathcal{R}_1$ has only 1 test $a$ with bias $\theta_a$. $\mathcal{R}_2$ has $T$ tests $\{b_1, \ldots, b_T\}$, each with bias $\theta_b$. The evaluation cost of each test is 1. The following condition is enforced

$$\theta_b^T = \theta_a + \varepsilon \tag{62}$$

Under such conditions, the MAXPROBREG algorithm would check tests in $\mathcal{R}_2$ before proceeding to $\mathcal{R}_1$. We compare the performance of this policy to the converse - one that evaluates $\mathcal{R}_1$ and then proceeds to $\mathcal{R}_2$.

Lets analyze the expected cost of MAXPROBREG. If $\mathcal{R}_2$ is valid, it incurs a cost of $T$, else it incurs a cost of $T + 1$. This equates to

$$\theta_b^T(T) + (1 - \theta_b^T)(T + 1)$$
$$= (T + 1) - \theta_b^T \tag{63}$$

We now analyze the converse which selects test $a$. If $\mathcal{R}_1$ is valid, it incurs a cost of 1, else it incurs $T + 1$. This equates to

$$\theta_a(1) + (1 - \theta_a)(T + 1)$$
$$= (T + 1) - \theta_a T \tag{64}$$

MAXPROBREG incurs a larger expected cost that equates to

$$(T + 1) - \theta_b^T - ((T + 1) - \theta_a T)$$
$$= \theta_a T - \theta_b^T \tag{65}$$
$$= \theta_a T - \theta_a - \varepsilon$$
$$= \theta_a(T - 1) - \varepsilon$$

$T$ can be made arbitrarily large to push this quantity higher.

We will now show that unconstrained BISECT will evaluate $\mathcal{R}_1$ in this case. We apply the BISECT selection rule in (14) to this problem. The utility of selecting test $a$ is

$$\Delta_{f_{\mathrm{DRD}}}(a \mid \mathbf{x}_\mathcal{A}) = (1 - \theta_a)(1 - \theta_b^T) - \left[\theta_a(1-1)(1 - \theta_b^T)\theta_a^2 + (1 - \theta_a)(1 - 0)(1 - \theta_b^T)(1 - \theta_a)^2\right]$$
$$= (1 - \theta_a)(1 - \theta_b^T) - (1 - \theta_a)^3(1 - \theta_b^T) \tag{66}$$

The utility of selecting test $b_i$ is

$$\Delta_{f_{\mathrm{DRD}}}(b_i \mid \mathbf{x}_\mathcal{A}) = (1 - \theta_a)(1 - \theta_b^T) - \left[\theta_b(1 - \theta_a)(1 - \theta_b^{T-1})\theta_b^2 + (1 - \theta_b)(1 - \theta_a)(1 - 0)(1 - \theta_b)^2\right]$$
$$= (1 - \theta_a)(1 - \theta_b^T) - \theta_b^3(1 - \theta_a)(1 - \theta_b^{T-1}) - (1 - \theta_b)^3(1 - \theta_a)$$
$$= (1 - \theta_a)(1 - \theta_b^T) - (1 - \theta_a)[\theta_b^3(1 - \theta_b^{T-1}) - (1 - \theta_b)^3] \tag{67}$$

We assume that $T$ is sufficiently large such that $\theta_b \approx 1$. Then the difference is

$$\Delta_{f_{\mathrm{DRD}}}(a \mid \mathbf{x}_\mathcal{A}) - \Delta_{f_{\mathrm{DRD}}}(b_i \mid \mathbf{x}_\mathcal{A}) = (1 - \theta_a)[\theta_b^3(1 - \theta_b^{T-1}) - (1 - \theta_b)^3] - (1 - \theta_a)^3(1 - \theta_b^T)$$
$$\approx (1 - \theta_a)[(1 - \theta_b^{T-1})] - (1 - \theta_a)^3(1 - \theta_b^T)$$
$$\approx (1 - \theta_a)(1 - \theta_b^T)[1 - (1 - \theta_a)^2]$$
$$\geq 0 \tag{68}$$

Hence unconstrained BISECT would significantly outperform MAXPROBREG in these problems. We now empirically show this result on a synthetic dataset as well as a carefully constructed 2D motion planning dataset. Table 3 shows a summary of these results.

Table 3: Normalized cost (95% C.I. lower / upper bound) with respect to unconstrained BISECT

| | MVOI | RANDOM Unconstrained MaxProbReg | MAXTALLY Unconstrained MaxProbReg | SETCOVER Unconstrained MaxProbReg | BISECT Unconstrained MaxProbReg |
|---|---|---|---|---|---|
| Synthetic | | $(6.50, 8.00)$ | $(5.50, 6.50)$ | $(3.00, 3.50)$ | $(0.00, 0.00)$ |
| ($T$ : 10) | $(3.00, 3.50)$ | $(3.00, 4.50)$ | $(5.00, 7.50)$ | $(3.00, 3.50)$ | $(3.00, 3.50)$ |
| 2D Plan | | $(9.50, 11.30)$ | $(2.80, 6.10)$ | $(6.60, 10.50)$ | $(0.00, 0.00)$ |
| ($m$ : 2) | $(6.60, 10.50)$ | $(6.90, 10.80)$ | $(6.80, 8.30)$ | $(6.60, 10.50)$ | $(7.30, 11.20)$ |
| 2D Plan | | $(2.44, 3.17)$ | $(2.83, 3.28)$ | $(2.50, 2.56)$ | $(0.00, 0.00)$ |
| ($m$ : 19) | $(0.89, 1.17)$ | $(1.06, 1.28)$ | $(0.89, 0.94)$ | $(0.78, 0.94)$ | $(0.78, 0.89)$ |

Figure 15: (a) The scenario where unconstrained BISECT outperforms MAXPROBREG significantly. (b) The 2D motion planning scenario where unconstrained BISECT outperforms others. Here the graph contains a straight line joining start and goal. With a low-probability, a block is placed between start and goal. This forces the path with many edges circumnavigating the block to have maximum probability. The straight line path joining start and goal has lower probability. Hence there is a similarity to the synthetic example in (a)

The first dataset is Synthetic which is a instantiation of the scenario shown in Fig. 15(a). We set $\theta_a = 0.9$, $\theta_b = 0.9906$, $\varepsilon = 0.01$, $T = 10$. We see MAXPROBREG does incurs 3 times more cost than the unconstrained variant.

The second dataset is a motion planning dataset as shown in Fig. 15(b). The dataset is created to closely resemble the synthetic dataset attributes. A RGG graph is created, and the straight line joining start and goal is added to the set of edges. A distribution of obstacles is created by placing a block with probability 0.3. The first dataset has 2 regions - the straight line containing one edge, and a path that goes around the block containing many more edges. Unconstrained BISECT evaluates the straight line first. MAXPROBREG evaluates the longer path first. We see MAXPROBREG does incurs 7 times more cost than the unconstrained variant.

The third dataset is same as the second, except the number of regions is increased to 19. Now we see that the contrast reduces. MAXPROBREG incurs 0.78 fraction more cost than unconstrained version.