[Reviews · NeurIPS 2017]

Reviewer 1



Overview and Summary This paper presents a method for motion planning where the cost of evaluating transitions between robot configurations is high. The problem is formulated as a graph-search algorithm, where the order of graph expansion has a large impact on the performance of the algorithm due to the edge expansion cost. The paper uses ideas from optimal test selection in order to derive the resulting algorithm. The algorithm is tested on a number of synthetic datasets, a simulator, and a real-world helicopter planning problem. Detailed Comments This paper extended work within the well-studied domain of robotic motion planning, extending and combining prior work in the area to construct a new algorithm. The idea of using optimal test selection to address the graph expansion ordering is a natural, and reasonable approach. The originality of this paper is moderate, and it does address a problem area of interest to many researchers. There are some issues with clarity in the paper. Specifically, starting in section 2, the term P(o) is first defined as an outcome vector, but then in equation (1) is used as a set to be drawn from (probabilistically?). Next, and more significantly, lines 129-130 define the set of subregions, consisting of a single consistent subregion and a number of disjoint subregions that are each invalid. The graph as a whole consists of nodes as subregions, and edge costs that are the product of the probabilities of each subregion. I can see two issues with this formulation that are not made clear within the context of the paper: - Given that the invalid subregions will expand with each test performed, the graph as a whole will have to change as testing progresses. - Because every edge contains at least one invalid subregion, every edge cost must contain at least one test where P(S_i) = 0, and therefore w must also equal 0. Due to this, either the problem formulation, or the clarity of the description must be changed. Also in terms of clarity, the index i is used on equations 3 and 5 on both the left and right hands of equations with different meanings. Finally, within equation 5, the last term of the top equation does not equate to the last term of the bottom equation. Within the results, the text states that there are 13 datasets, but the results table lists 14. Line 215 states that there are 8 tests with exclusively best performance, but the table lists only 7. The claim on line 218 of a factor of 20 improvement is not possible to verify due to the normalization of all results. Finally, and most importantly, the results are not compared against any of the other motion planning algorithms discussed in the related work, only against self-constructed heuristics used within the same general framework. We have no idea how these algorithms compare to the state of the art. Post-feedback: Thanks to the authors for their explanation, additional material, and commitments to update the paper. In light of this interaction, I have modified my score.

Reviewer 2



The paper presents an active learning algorithm for solving the Decision Region Determination problem (DRD) when the hypothesis space is determined by bernoullii independent outcomes. The motivation comes from robotic planning, but both the DRD problem and the Bernoulli assumption are interesting in their own right. In DRD, as in the classic active learning setting, a hypothesis (i.e., map from a query space to a binary label/outcome space) is sampled from a known prior distribution. In contrast to classic active learning, where the objective is to learn the hypothesis with as few queries as possible, the active learner is given a set of (possibly overlapping) regions, where each region is a subset of the query space. The objective is then to identify a region that in which all queries are labeled true by the correct hypothesis as quickly as possible. An algorithm with provable guarantees for DRD already exists from [4], it self an extension of the EC2 algorithm from [16]. When the prior is a multinomial (i.e., outcomes of every query are bernoulli with known parameters), the algorithm requires an exponential computation to determine the next query: the authors show that a simple modification along with a memoization trick makes this linear, while still maintaining adaptive submodularity (the property that yields the optimality guarantees). Additional modifications are provided at different steps of the algorithm, that lead to further improvements in query complexity. The algorithm is evaluated against a barrage of state-of-the-art (non-strawmen) competitors on both synthetic and real-life datasets. This paper is overall well written and lucid despite the technical depth, though familiarity with [4] and [16] would help. The contribution is a bit incremental compared to [4], but there are clear differences, and the algorithms constructed go beyond what was possible through [4]. Moreover, multinomial prior is a reasonable model in the absence of additional structure, so having an algorithm under which this is tractable is important and deserves to be reported at NIPS. The authors clearly explain how their work relates to [4][16], and the experiments are thorough and exhaustive Minor comments: Line 61: move P(o) to right after the word "distribution" (it is the prior, I am guessing). The motivation of the problem goes beyond robot planning; that said, if this is to be used in the introduction as the major problem solved, citations should not be limited to [10,11,8]: motion planning is classic, and dynamic programming and MDPs must be mentioned as alternatives (though their objective is not feasibility/validity).

Reviewer 3



This paper considers the problem planning in binomial graphs in which evaluating the edges is expensive, which can be used for planning collision-free paths for robots. Naive planning takes exponential time. This papers links motion planning with Bayesian active learning to select a feasible path. By building off of prior work and leveraging the binomial edge assumption, they develop an efficient, near-optimal algorithm. Finally, experiments on multiple simulated and real domains show the efficacy of the approach compared to prior, non-trivial baselines. The contributions are: (1) Bisect, which computes each subproblem in linear time (as opposed to exponential) (2) A novel method for selecting candidate tests Overall, the paper presents the motivation, contributions, and experiments effectively and convincingly, and therefore I think the paper should be accepted. More detailed comments below: (a) The technical section seems to require knowledge of prior work, making it difficult for me to evaluate the technical details, however due to page limit constraints there is not much that can be done. (b) The results showing that MaxProbReg candidate test selection usually is responsible for most of the speed improvement compared to using Bisect makes me as a reader question if it’s worth my trouble to implement Bisect. However, the experiments showing the cases where Bisect helps a lot is good; what would be more convincing is showing the reader that these problems do appear in the real-world (which the current real-world experiments do not demonstrate). However, because the authors will be open-sourcing their code, the issue of implementing Bisect is less problematic.